# Characterisation of particle single scattering albedo with a modified airborne dual-wavelengths CAPS monitor

Chenjie Yu[1], Edouard Pangui[2], Kevin Tu[2], Mathieu Cazaunau[2], Maxime Feingesicht[2], Landsheere Xavier[2], Thierry Bourrianne[3], Vincent Michoud[1], Christopher Cantrell[2], Timothy B. Onasch[4], Andrew Freedman[4], and Paola Formenti[1]

[1] Université Paris Cité and Univ Paris Est Créteil, CNRS, LISA, F-75013 Paris, France
[2] Univ Paris Est Créteil and Université Paris Cité, CNRS, LISA, F-94010 Créteil, France
[3] CNRM, CNRS/Météo France, Toulouse, France
[4] Aerodyne Research Inc., Billerica, MA 01821-3976, USA

*Correspondence to*: Chenjie Yu (chenjie.yu@lisa.ipsl.fr) and Paola Formenti (paola.formenti@lisa.ipsl.fr)

**Abstract**

Atmospheric aerosols impact the Earth's climate system directly by scattering and absorbing solar radiation, and it is important to characterise the aerosol optical properties in detail. This study reports the development and validation of an airborne dual-wavelength cavity-attenuated phase shift-single (CAPS) monitor, named A2S2 (Aerosol Absorption Spectral Sizer) based on the commercial CAPS single scattering albedo monitor (CAPS-PM$_{SSA}$, Aerodyne), to simultaneously measure the aerosol optical scattering and extinction at both 450 nm and 630 nm wavelengths. Replaced pressure and temperature sensors and an additional flow control system were incorporated into the A2S2 for its utilization onboard research aircraft measuring within the troposphere. The evaluation of A2S2 characteristics was performed in the laboratory and included the investigation of the signal-to-noise ratio, validation of performance at various pressure levels, optical-closure studies and intercomparing with the currently validated techniques. The chamber experiments show that the A2S2 can perform measurements at sample pressures as low as 550 hPa and at sample temperatures as high as 315K. Based on the Allan analysis results, we have evaluated that the minimum detection limit of the measurements is  show that the measurements are with a limit accuracy of ~2 Mm$^{-1}$ at 450 nm and ~1 Mm$^{-1}$ at 630 nm for 1 Hz measurements of both scattering coefficients ($\sigma_{sca}$) and extinction coefficients ($\sigma_{ext}$). The optical-closure study with size-selected polystyrene latex (PSL) particles show that the truncation error of the A2S2 is negligible for particles with particle volume diameter ($D_p$) < 200 nm, while for the larger sub-micrometre particles, the measurement uncertainty of A2S2 increases but remains less than 20%. The average factors to correct the truncation error are 1.13 and 1.05 for 450 nm and 630 nm, respectively. A simplified truncation correction, dependent on the Scattering Ångström Exponent (SAE), was developed to rectify truncation errors of the future A2S2 field measurements data. The $\sigma_{ext}$ and $\sigma_{ext}$ measured by A2S2 shows good agreement with the concurrent measured results from the nephelometer and the CAPS-PM$_{ex}$ (Particle Extinction Monitor). The absorption coefficient $\sigma_{abs}$ derived through the extinction-minus-scattering (EMS) method by the A2S2 also corresponds with the results obtained from the aethalometer. The A2S2 was successfully deployed during an aircraft measurement campaign (ACROSS) conducted in the vicinity of Paris and the surrounding regions. The average SSA measured during the entire ACROSS flight campaign is 0.86 and 0.88 at 450 nm and 630 nm, respectively, suggesting that

light-absorbing organic aerosols play a significant role. The average SAE and Absorption Ångström Exponent (AAE) varied
due to measurements in various pollution conditions. The results presented in this study indicate that the A2S2 instrument is
reliable for measuring aerosol $\sigma_{sca}$ and $\sigma_{ext}$ at both blue and red wavelengths, and it stands as a viable substitute for future
airborne evaluations of aerosol optical properties.

**Introduction**

Atmospheric aerosols, particularly light-absorbing carbonaceous aerosols and mineral dust, play a significant role in global
radiative transfer by scattering and absorbing solar radiation directly, a phenomenon referred to as the direct aerosol effect
(Jacobson, 2012; Riemer et al., 2019; Liu et al., 2020). It is known that the radiative forcing impact of aerosols is mainly driven
by three important parameters (Haywood and Shine, 1995): the aerosol optical depth (AOD), the single scattering albedo (SSA)
and the asymmetry parameter (*g*). The AOD is the integration of extinction coefficients over a certain path-length, and it
represents the optically active concentration fields. The SSA is derived from the ratio between the scattering and extinction.
As it quantifies the fraction of the incoming light that is scattered by a particle or substance compared to the portion that gets
absorbed, the SSA is the key parameter to determine the overall uncertainty in aerosol direct and semi-direct effects. The *g*
parameter quantifies the preferential directions of light photons that are scattered by particles.

To obtain the aerosol optical properties, various measurements have been conducted by satellites and suborbital instruments
in recent decades. Suborbital measurement mainly encompasses airborne and ground-based in-situ and remote sensing
measurements. At present, Earth-orbiting satellite networks (e.g., MODIS) provide comprehensive global coverage of AOD
distributions. But the capability of satellites to acquire quantitative aerosol optical properties, specifically the spectral
dependence of SSA, is still limited and the need is evident for new intensive airborne measurements to constrain the aerosol
microphysical properties assumption and vertical structure to improve space-based remote sensing retrieval algorithms (Peers
et al., 2019; Kahn et al., 2023). Various in situ techniques exist to derive different aerosol optical properties. For the absorption
coefficient ($\sigma_{abs}$) measurements, the filter-based technique is commonly employed by online measurement instruments such
as the aethalometer (Hansen et al., 1984) (e.g. AE33, Magee Scientific used in this study (Drinovec et al., 2015)), the Particle
Soot Absorption Photometer (PSAP, Radiance Research) (Bond et al., 1999), the Multi Angle Absorption Photometer (MAAP,
Thermo Scientific) (Petzold and Schönlinner, 2004) and the Tricolor Absorption Photometer (TAP/CLAP) (Ogren et al., 2017).
In the filter-based technique, light transmittance of a filter is continuously monitored, and the $\sigma_{abs}$ is derived through the
transmittance changes caused by particles deposited onto the filter. A major disadvantage of this method is the non-negligible
multi-scattering effect of filter material and the deposited particles, and this issue is related to several factors including relative
aerosol loading, humidity, and SSA (Moosmüller et al., 2009). For example, the filter-based absorption measurement method
utilized by PSAP requires corrections to account for alterations in both scattering aerosol loading and aerosol transmissions
(Virkkula et al., 2005; Virkkula, 2010). Moreover, the relatively slow measurement frequency of the filter-based measurement

techniques makes them not ideal for the airborne measurements due to their slow time resolution, especially during altitude profiles. The scattering coefficient ($\sigma_{sca}$), is commonly characterised by the nephelometry technique. The nephelometer analyses the particle scattering intensity collected in a wide but limited range of scattering angles (7°- 170°), causing the loss of near forward and near backward scattering characterisation, a phenomenon commonly referred to as the truncation error

(Heintzenberg and Charlson, 1996). Technological advancements have allowed more precise direct measurements of the aerosol extinction coefficient ($\sigma_{ext}$). The extinction coefficient can be characterised by cavity ring-down spectroscopy (CRD) (Moosmüller et al., 2005; Baynard et al., 2007),  Cavity Attenuated Phase-Shift (CAPS) (Kebabian et al., 2005; Kebabian et al., 2007) and sun photometry (Karol et al., 2013; Schmid et al., 2003).

In this study, we focus on the development and deployment of a CAPS-based instrument for airborne applications. The CAPS-based instrument employs a light emitting diode (LED) as its light source, and the $\sigma_{ext}$ is derived by quantifying the variations in the phase shift of the distorted waveform caused by the modulated light passing through a highly reflective optical cell. Compared to the custom-built CRD-based instrument, the CAPS-based instrument is compact and robust. The Cavity Attenuated Phase-Shift Particle Extinction Monitor (CAPS-PM$_{ex}$) instrument (Massoli et al., 2010), developed by Aerodyne

Inc, utilizes the CAPS technique to enable highly sensitive in-situ measurements of the extinction coefficient. Based on the same CAPS technique, Aerodyne Inc. introduced the CAPS single scattering monitor (CAPS-PM$_{SSA}$) to derive both the extinction and scattering measurements in the same sample cell (Onasch et al., 2015). The CAPS-PM$_{SSA}$ incorporates an integrating sphere, which minimize the bias of the light collections with respect to angle when measuring $\sigma_{sca}$, and the $\sigma_{abs}$ can also be derived indirectly through the extinction-minus-scattering (EMS) method. Compared to the measurements obtained

by combining separate instruments (e.g., one nephelometer for $\sigma_{sca}$ and one filter-based instrument for $\sigma_{abs}$), the CAPS-PM$_{SSA}$ offers distinct advantages as there is no need to employ different time or wavelength averaging, or inlet differences into consideration to derive the aerosol optical properties. The application of CAPS-PM$_{SSA}$ makes significant progress in the characterisation of aerosol optical properties characterisation, and the CAPS-PM$_{SSA}$ has been deployed in several different laboratory and ground-based ambient measurement studies (Han et al., 2017; Zhao et al., 2017; Corbin et al., 2018; Corbin et

al., 2020; Corbin et al., 2022).

The properties including fast response and compact size, make CAPS-PM$_{SSA}$ an ideal instrument for the airborne measurements of aerosol optical properties. Nevertheless, to address the requirements of aerosol optical properties measurement from airborne platforms, an improved flow control system is required to maintain the instrument flows under the reduced-pressure

conditions that are common during the airborne measurements. In addition, a crucial requirement is to conduct dual-wavelength measurements within the same sample volume, and thus a redesigned inlet was required. In this paper, we describe the modification and characterisation of a new airborne dual CAPS-PM$_{SSA}$ (450 nm and 630 nm) measurement system, the Aerosol Absorption Spectral Sizer (A2S2), and the validation of its performance through laboratory experiments and in-situ airborne measurements within the area around Paris (Île-de-France). The A2S2 measurements are compared to the results from

performance validated instruments. By providing with vertical profiles of climate-relevant properties such as aerosol single scattering albedo and aerosol optical depth, the A2S2 measurement results can play a role in helping the evaluation of the direct and semi-direct radiative effects in modelling studies and contribute to future advancements and validation of aerosol remote sensing products (Formenti et al., 2018).

## 2 Instruments and experiment methods

### 2.1 CAPS-PM$_{SSA}$

The design of the CAPS-PM$_{SSA}$ is described in Onasch et al. (2015), and the diagram of CAPS-PM$_{SSA}$ is presented in Fig.1 (a). Briefly, the CAPS-based technique consists of two high reflectivity mirrors (~ 99.99% reflectivity) within the sample cell, and this configuration allows for a long effective optical path length. An Aerodyne manufactured LED light is used as the input light source and is available at multi wavelengths ranging from 450 nm to 780 nm. The customed LED input light is square-wave modulated (typically at 17kHZ), and the detected waveform is distorted and exhibits a phase shift at the fundamental frequency of the initial modulation. This phase shift is related to various factors including the instrument geometric properties and the presence of optically active aerosol particles. Hence, the $\sigma_{ext}$ is determined as changes in the phase shift between the measurement when particles introduced into the optical cavity and the particle-free baseline measurement. Shown in Fig 1, an additional integrating sphere with an inner diameter of 10 cm is used to characterise the $\sigma_{sca}$. The inner surface of the sphere is coated with highly-reflective material and shows a Lambertian reflectivity efficiency of 98%. A photomultiplier tube (PMT, Hamamatsu) module is then used to sample the scattered light and output the signal for further processing. The integrating spherical design helps maximise the PMT collection efficiency of scattered light and reduce the measurement bias related to truncation angles.


### 2.2 Aerosol Absorption spectral sizer (A2S2)

To achieve dual-wavelength measurements of aerosol extinction and scattering at the same time and with airborne capabilities, we integrated two CAPS-PM$_{SSA}$ sample cells (450 and 630 nm, respectively) into a single measurement package that is designated A2S2. The diagram of the A2S2 is shown in Fig 2. The inlet has been redesigned to meet the requirements of dual-wavelength measurements within the same sample volume, and the particle loss rate for the modified inlet system is estimated to be less than10% for the particles with diameters up to 4 μm using the simulation method of Von Der Weiden et al. (2009). The flow system has been modified by incorporating a separated sampling pump into the system, and it provides a constant sample flow rate of ~1.7 litre per minute (L min$^{-1}$) (~0.85 L min$^{-1}$ for each cell) which is regulated by the critical orifice within each sample cell. A three-way solenoid valve was placed upstream of each sampling cell to enable the instrument to switch between baseline mode and sampling mode. The purge flow is generated by the same diaphragm pump as the original CAPS-

PM$_{SSA}$, providing a continuous flow at a rate of 0.025 L min$^{-1}$ also regulated by critical orifices which serve to prevent the high-reflectivity mirrors from contamination by deposited particles. Existing temperature and pressure sensors of CAPS-PM$_{SSA}$ were replaced with new temperature (DS18B20 by Maxim Integrated) and pressure (A-10-12719316 by WIKA) sensors, ensuring accurate monitoring of temperature and pressure to detect any leaking during airborne measurements. The new pressure sensor has a measurement range from 0 to 40,000 hPa (within 0.5% uncertainties) while the range for the new temperature sensor is -55 °C – 125 °C ($\pm$0.5 °C). The performance validation tests of the new pressure sensor are presented in the supplemental section. A custom software interface was developed to control the entire A2S2 system and output the instrument data that includes the sensors. The response time of A2S2 is 1 Hz, and it is programmed to carry out a continuous measurement phase for 2 min, which will be succeeded by a period of 1 min dedicated to cell flushing and establishing the baseline characteristics.

## 2.2 Laboratory validation

The laboratory performance validation of A2S2 was performed at the Laboratoire Interuniversitaire des Systèmes Atmosphériques (LISA) in Creteil, France. These tests include the characterisation of signal-to-noise ratio, the performance under reduced pressure and elevated temperature conditions, the angular truncation, and intercomparison with the other optical measurement instruments.

The signal-to-noise ratio was tested by conducting continuous measurements of aerosol-free air for several hours, and this was achieved by sampling the A2S2 though a HEPA filter (TSI). Auto baseline characterization was disabled throughout the characterization period. Then the Allan variance (Allan, 1966) was determined to assess the stability and noise characteristics of the measurements over different averaging time scales. Due to the potential lower aerosol loading from the airborne measurements compared to the ground-based measurements, the Allan variance approach is useful to assess the stability of the A2S2 as modified. The Allan variance is also helpful for selecting the appropriate data filtering or processing approaches to improve the measurement precision.

To validate the performance of the A2S2 under low-pressure conditions at high altitudes and the potentially high temperature environment within the cabin during low altitude airborne measurements, we conducted an intercomparison study at controlled pressure and temperature levels. This involved connecting both the modified CAPS-PM$_{SSA}$ at 630 nm sampling cell and the original CAPS-PM$_{ex}$ at 630 nm (Aerodyne Inc) to the CESAM (Multiphase Atmospheric Experimental Simulation Chamber) chamber. The details of the CESAM chamber facility are described in Wang et al. (2011). The configuration of the experiment is presented in the supplement. The experiment starts with the addition of ammonium sulfate particles (~250 μg) into the chamber that is at standard pressure (1013.25 hPa), and then the pressure within the chamber was pumped to decrease the pressure stepwise to 900 hPa, 800 hPa, 700 hPa, 550 hPa, 400 hPa and 200 hPa. Each pressure level was maintained for least

~30 min. In addition, the CAPS-PM$_{SSA}$ 630 nm was placed in a temperature-controlled box, and the temperature was set to increase gradually from ~300 K (~26.8 ℃) to ~315 K (~41.9 ℃) to simulate the high temperature condition within the cabin, while the CAPS-PM$_{ex}$ was exposed to the ambient atmosphere. In this experiment, both the modified CAPS-PM$_{SSA}$ 630 nm and the CAPS-PM$_{ex}$ 630 nm were set to perform a 12-min duty cycle which includes 10-min measurements and 2-min flushing and baseline characterisation. The results from both CAPS-PM$_{SSA}$ and CAPS-PM$_{ex}$ have been corrected to standard temperature (273.15 K) and pressure and (1013.25 hPa) conditions using measurements from the modified pressure and temperature sensors for intercomparison. The Due to the pumping of the sampling instrument, and the chamber dilution is corrected following the description described in Lamkaddam et al. (2017):

$$\sigma_{\text{corrected}}(t_{i+1}) = \sigma_{\text{corrected}}(t_i) + \Delta\sigma_{\text{measured}} + \frac{Q_{\text{p}} \times \Delta t}{v}\sigma_{\text{measured}}(t_i)e^{-\frac{Q_{\text{p}} \times \Delta t}{v}} \quad (1)$$

Where $\sigma_{\text{corrected}}(t)$ is the dilution corrected coefficient at time $t$, $\sigma_{\text{measured}}(t)$ is the measured coefficient at time $t$, $\Delta\sigma_{\text{measured}}(\Delta t)$ is the change of measured coefficient over time $\Delta t$, and $v$ is the CESAM chamber volume (4200 L), and $Q_{\text{p}}$ is the total flow rate of CESAM chamber.

The angular truncation error of the A2S2 is quantified by comparing the measured scattering coefficient with the scattering coefficient derived from Mie theory calculations. The configuration of the truncation characterisation experiment is shown in Fig 3(a). Nebulised and dried PSL spheres with standard particle volume equivalent diameters ($D_{\text{p}}$) of 200, 350, 500, 600 and 800 nm were selected by an Aerodynamic Aerosol Classifier (AAC, Cambustion). The schematic and validation of the AAC is described in a previous publication (Tavakoli and Olfert, 2013). The AAC can generate monodisperse distributions of particles based on their aerodynamic sizes according to particle relaxation time without needing charging electrostatic elements. In contrast to electrostatic aerosol classifiers such as the differential mobility analyzer (DMA), the AAC can provide monodisperse results that are less affected by particle compositions, morphologies, and sizes. The aerodynamic diameter of the PSL particles is converted to volume diameter following the methods described in previous publications (Decarlo et al., 2004; Yu et al., 2022). The particle density and the shape factor of PSL particles were determined to be 1.05 g/cm³ and 1 (perfect sphere), respectively, and the refractive index of PSL particles is 1.59 + 0$i$. A Condensation Particle Counter (CPC, TSI 3775) and the A2S2 were placed downstream of the AAC, and the CPC was used to record the total PSL particle number concentration at each AAC-selected size point. The scattering efficiency at 450 nm and 630 nm over all the angles (0° - 180°) at selected $D_{\text{p}}$ ($Q_{\text{sca}}^{\text{Mie}}(D_{\text{p}})$) is calculated by Mie theory for spherical homogeneous particles following the methods described by Bohren and Huffman (1983). The measured truncation error of the A2S2 is defined as the ratio between the scattering efficiency measured ($Q_{\text{sca}}^{\text{A2S2}}(D_{\text{p}})$) and that calculated from Mie theory $Q_{\text{sca}}^{\text{Mie}}(D_{\text{p}})$:

$$trunc(D_{\mathrm{p}})_{\mathrm{measure}} = \frac{Q_{\mathrm{sca}}^{\mathrm{A2S2}}(D_{\mathrm{p}})}{Q_{\mathrm{sca}}^{\mathrm{Mie}}(D_{\mathrm{p}})} = \frac{\sigma_{\mathrm{sca}}^{\mathrm{A2S2}}/\frac{\pi}{4}\cdot D_{\mathrm{p}}^2 \cdot N}{Q_{\mathrm{sca}}^{\mathrm{Mie}}(D_{\mathrm{p}})} \qquad (2)$$

Where $N$ is the average number concentration over the sampling period measured by CPC.

Intercomparison of A2S2 with the Nephelometer (NEPH, TSI 3563), two CAPS-PM$_{\mathrm{ex}}$ (450 nm and 630 nm, Aerodyne Inc.), and Aethalometer-33 (AE33, Magee Scientific) was performed using the nebulised standard particles. In addition, a Scanning Mobility Particle Sizer (SMPS, TSI) was used for the aerosol size distribution measurements. The SMPS comprised a DMA (TSI 3081) and a CPC (TSI 3775). The detailed list of the intercomparison instruments and the correction method references are presented in Table 1, and the setting of the intercomparison experiments is shown in Fig 3(b). Briefly, the NEPH measures scattering coefficient at 450 nm, 550 nm and 700 nm, the AE33 characterises the aerosol absorption coefficient at 7 wavelengths ranged from 370 nm to 950 nm, and the two CAPS-PM$_{\mathrm{ex}}$ measure the $\sigma_{\mathrm{ext}}$ at 450 nm and 630 nm. Three case studies were conducted including pure ammonium sulfate (99.99%, Merck KGaA), pure Aquadag (Aqueous Deflocculated Acheson Graphite, Acheson Industries Inc.), and an external mixture of Aquadag and ammonium sulfate. Each sampling period had constant $\sigma_{\mathrm{ext}}$ and $\sigma_{\mathrm{sca}}$ levels and was measured for 10 min. The NEPH was calibrated with $CO_2$ before the lab experiments, and the truncation error of NEPH was corrected following the correction algorithm described in Anderson and Ogren (1998). The multiple-scattering correction factor of the AE33 was determined following the polar photometer approach factor introduced by Bernardoni et al. (2021). For comparisons at the appropriate wavelengths, the NEPH and AE33 results have been scaled using the Ångström exponent approach using equation (3):

$$xAE = -\frac{\ln(\sigma_{\lambda_1}/\sigma_{\lambda_2})}{\ln(\lambda_1/\lambda_2)} \qquad (3)$$

Where $xAE$ is the Scattering, Absorption, or Extinction Ångström Exponent (SAE, AAE, EAE), $\sigma_{\lambda_1}$ and $\sigma_{\lambda_2}$ represent the scattering, absorption, or extinction coefficient at wavelengths $\lambda_1$ and $\lambda_2$ respectively. The absorption or the scattering coefficient ($\sigma_\lambda$) at a given wavelength ($\lambda$) can be derived through equation (4):

$$\sigma_\lambda = \sigma_{\lambda_0} \cdot \left(\frac{\lambda}{\lambda_0}\right)^{-xAE} \qquad (4)$$

Where $\sigma_{\lambda_0}$ is the absorption or scattering coefficient at the wavelength $\lambda_0$. In this study the scattering coefficient at 630 nm for the NEPH is derived through measurements at 700 nm, and the absorption coefficient at 450 nm and 630 nm for the AE33 is derived through absorption measurements at 470 nm and 660 nm, respectively.

The measurement uncertainty is also listed in Table 1. For the A2S2, the uncertainty of the $\sigma_{abs}$ derived from EMS method is 13% according to Onasch et al. (2015) and Pfeifer et al. (2020). The uncertainties for the angström exponent are derived through the Gaussian error propagations (Weber et al., 2022):

$$\Delta xAE = \sqrt{(\frac{-1}{\ln(\lambda_1/\lambda_2)\cdot\sigma_{\lambda_1}} \cdot \Delta\sigma_{\lambda_1} \cdot \sigma_{\lambda_1})^2 + (\frac{1}{\ln(\lambda_1/\lambda_2)\cdot\sigma_{\lambda_2}}\Delta\sigma_{\lambda_2} \cdot \sigma_{\lambda_2})^2} \qquad (5)$$

Where $xAE$ represents EAE, AAE or SAE; $\Delta\sigma$ represents the measurement uncertainty of the extinction, absorption and scattering coefficient measurement at certain wavelength.

## 2.3 Airborne measurements

The French environmental research aircraft ATR-42 managed by SAFIRE (Service des Avions Français Instruments pour la Recherche en Environnement) was used to sample urban pollution as part of the ACROSS (Atmospheric ChemistRy Of the Suburban foreSt) project (Cantrell and Michoud, 2022). Airborne measurements were performed between 13th June and 7th July, 2022 over the Paris suburban areas (Île-de-France) and surrounding regions, as presented in Fig. 4. Measurements were performed mostly within the boundary layer with an altitude around 300 m above ground level (a.g.l). Altitude profile measurements were carried out by ascending to ~3500 m a.g.l. on June 18th, 21st, 23rd and 27th.

Onboard the aircraft, both the A2S2 and the NEPH was connected to the AVIRAD measurement system. The AVIRAD system consists of an isoaxial and isokinetic inlet which has a collection efficiency of 50% for particles with 12 μm optical diameters (Formenti et al. (2011), with various sampling instruments are connected to the inlet. The AVIRAD has been deployed on multiple airborne projects including dust events and pollution characterisations (Di Biagio et al., 2015; Di Biagio et al., 2016). The NEPH was calibrated with $CO_2$ and corrected for truncation error through the methods described by Anderson and Ogren (1998). Presented later in Fig. 10, the average and median SSA during the ACROSS airborne measurement period exceeded 0.7. Massoli et al. (2009) indicates that the uncertainties associated with applying the truncation correction method, as outlined by Anderson and Ogren (1998), to NEPH scattering are within 5% when the SSA is greater than 0.7. Due to the complex configuration of the spherical nephelometer within the A2S2, it is challenging to apply the conventional truncation correction approaches (Modini et al., 2021). As an alternative, the A2S2 is corrected based on the average truncation characterisation results obtained in the lab. To enhance the signal-to-noise ratio, the data has been averaged over 10s for all the flights, and all the data has been corrected to standard temperature (273.15 K) and pressure and (1013.25 hpa) for intercomparison. Before the airborne measurement experiments, the scattering channels of A2S2 were calibrated by nebulised polystyrene latex (PSL) spheres 200 nm (SSA = 1) following the normal CAPS-PM$_{SSA}$ calibration procedure.

# 3 Results

## 3.1 Laboratory instrument validation results

### 3.1.1 Signal-to-noise ratio

The Allan variance analysis is presented in Fig. 5. The Allan standard deviation for the 450 nm extinction measurement increases with integration time after 40s. This is due to the $\sigma_{ext}$ baseline is assumed to remain constant by the A2S2 over the signal-to-noise experiment period, but the actual contribution of the ambient gas phase species (mainly $NO_2$) absorption to the total extinction varies with extinction time (Massoli et al., 2010). The uncertainty should be less during ambient measurements if the baseline is characterised more frequently than once per hour in operation. Nonetheless, the previous study by Pfeifer et al. (2020) shows that the variation of the gas phase $\sigma_{ext}$ baseline at 450 nm for the CAPS-PM$_{ex}$ may lead to an uncertainty up to around 0.8 Mm$^{-1}$ min$^{-1}$ for the ambient $\sigma_{ext}$ characterisations. To minimise the uncertainty of the baseline variation in CAPS-based instruments, a frequent baseline characterisation is needed. In this study, measurements were conducted at locations distant from the emission sources, and baseline values were measured every 2 min to reduce the influence from the background signal. The Allan standard deviation for the 630 nm extinction and scattering measurements are smaller compared to the 450 nm measurements, and the extinction measurement at 630 nm is less influenced by baseline drift issues. The minimum detection limit (MDL) involves a calculation derived from three times the Allan standard deviation, and the detection limit at an integration time of 1s, 10s and 30s is presented in Table 3. Our laboratory results show that, at a measurement frequency of 1 Hz, the MDL is 1.89 Mm$^{-1}$ for $\sigma_{ext}$ and 2.25 Mm$^{-1}$ for $\sigma_{sca}$ at 450 nm. At 630 nm, the MDL is 0.69 Mm$^{-1}$ for $\sigma_{ext}$ and 1.08 Mm$^{-1}$ for $\sigma_{sca}$. With the increase of integration time to 10s, the limits are reduced to 0.69 Mm$^{-1}$ and 0.21 Mm$^{-1}$ for $\sigma_{ext}$ at 450 and 630 nm, respectively, and to 0.45 Mm$^{-1}$ and 0.12 Mm$^{-1}$ for $\sigma_{sca}$ at 450 and 630 nm, respectively. The Allan analysis indicates that the limit of detection reaches its minimum value at an integration time of 30s. Nevertheless, in the case of ACROSS airborne measurements, there is a requirement for high-frequency results to effectively characterize aerosol optical properties, especially during instances when the research aircraft intercepted with urban plumes or conducted altitude profile measurements. The minimum $\sigma_{ext}$ and $\sigma_{sca}$ observed during ACROSS campaign is both ~1.35 Mm$^{-1}$ at 450 nm, and both ~0.32 Mm$^{-1}$ at 630 nm. Hence, the detection limit achieved with a 10s integration time is deemed satisfactory to fulfill the requirements of the ACROSS project measurement.

### 3.1.2 Performance under simulated low-pressure environment

Fig. 6 presents the results of the chamber measurements made at various controlled pressure levels, and the temperature of the modified CAPS-PM$_{SSA}$ increased slowly to 315K after the injection of the ammonium sulfate. The dilution and STP corrected $\sigma_{ext}$ measured by the original CAPS-PM$_{ex}$ unit agrees well with the $\sigma_{ext}$ from the modified CAPS-PM$_{SSA}$ at a constant pressure of ~1013.25 hPa and ~900 hPa. However, our original CAPS-PM$_{ex}$ unit is unable to deliver an accurate measurement when the pressure within the chamber drops to 800 hPa or less. The $\sigma_{ext}$ and $\sigma_{sca}$ reported by the modified CAPS-PM$_{SSA}$ showed minimal impact until the pressure reached ~550 hPa. When the pressure drops further to ~400 hPa, the signal noise level

increases. The chamber experiment results validate that our modification to the CAPS-PM$_{SSA}$ can provide accurate measurements with ambient pressures as low as 550 hPa and instrument temperatures as high as 315K.

**3.1.2 Angular truncation characterisation and correction**

Fig. 7 presents the data collected to determine the truncation of the A2S2 instrument at 450 nm and 630 nm wavelengths. The truncation measured and simulated by Onasch et al. (2015) and Modini et al. (2021) are also included for comparison. Compared to the simulation reported by Onasch et al. (2015) (MieAmigo), Modini et al. (2021) accounts for the reflection of scattering light from the inner surface of the glass sampling tube within the integrating nephelometer, and this reflection phenomenon is simulated for a path length range of 0 to 4.7 cm. Hence the two simulation methods are referred as simulation

with and without reflection. At both 450 nm and 630 nm wavelengths, the AAC-selected PSL particle results show that the truncation for particles with $D_p$ up to 200 nm is insignificant, and the truncation uncertainty is less than 10% for particles with $D_p$ up to 400 nm. For larger submicron particle size ($D_p$ between 400 nm and 1000 nm), the truncation of 630 nm wavelengths is around 10% while the truncation of 450 nm is greater and is around 20%. This observation is consistent with observations in the Rayleigh scattering regime, where larger particles exhibit near-forward scattering that is not captured by the CAPS-

PM$_{SSA}$ monitors. The average truncation error for particles with $D_p$ between 200 nm and 1000 nm is 0.86 and 0.94 at 450 nm and 630 nm, respectively.

Compared to the truncation results for the CAPS-PM$_{SSA}$ presented in previous studies, the truncation results of A2S2 in this study at 450 nm and 630 nm wavelength are greater than the values reported by Onasch et al. (2015) but are closer to the values

reported by Modini et al. (2021). Modini et al. (2021) suggested that the experiments done by Onasch et al. (2015) may be affected by multiply-charged particles, while the AAC source is not influenced by the multi-charging issues since particles are sized by an aerodynamic method. Another possible explanation for the differences could be the slight variations in the configurations of CAPS-PM$_{SSA}$ sample cells from one instrument to another, and our truncation results presented in this study may solely reflect the potential measurement error of our A2S2. The simulated truncation from different methods is also

presented in Fig 7. The truncation simulation of Onasch et al. (2015) shows the smallest correction, which is less than 10%. However, the simulation that includes reflection done by Modini et al. (2021) shows a larger truncation correction of around 15%. Though the simulation results of Modini et al. (2021) indicate that the self-reflection of the sampling tube may be another source of the uncertainty, there is no clear evidence that this will lead to significant measurement error and the largest uncertainty is expected to arise from truncation itself. Overall, our truncation experiment results show a trend similar to the

simulated results. The findings indicate that the A2S2 is less affected by truncation for the fine mode particle measurements. But for the studies where contributions from coarse mode particles are present, larger measurement errors from the A2S2 are expected, especially for the 450 nm wavelength.

Based on the characterisation results reported here, we introduce a simplified correction algorithm as a function of measured uncorrected Scattering Ångström Exponent (SAE) to apply to ambient measurement results. The correction function is presented in Fig. 7(c). Derived from the average truncation calculated above, the average correction factors are 1.13 and 1.05 at the wavelengths of 450 nm and 630 nm, respectively. Subsequently, the truncation is corrected based on the time-resolved measured uncorrected SAE between 450 nm and 630 nm observed by A2S2: when the SAE falls below 1, indicating the dominance of larger particles, the correction function is applied to the measurement results. Conversely, in situations dominated by fine particles (SAE > 1), there is no need to apply the correction due to the minimal truncation observed during characterisation experiments.

### 3.1.3 Instruments intercomparison

Fig. 8 shows comparisons between A2S2 and the performance validated instruments. Panels (a) and (b) are for the CAPS-PM$_{ex}$ ($\sigma_{ext}$), panels (c) and (d) use the NEPH ($\sigma_{sca}$) and panels (e) and (f) use the AE33 ($\sigma_{abs}$). Each point shown in Fig. 8 represents the average value computed over each measurement period with constant conditions. For the experiments with pure ammonium sulfate and pure Aquadag, the intercomparisons are performed under high (> 200 Mm$^{-1}$ at 450 nm, and > 150 Mm$^{-1}$ at 630 nm), moderate (~100 Mm$^{-1}$ - ~200 Mm$^{-1}$ at 450 nm, and ~50 Mm$^{-1}$ - ~150 Mm$^{-1}$ at 630 nm), and low (~50 Mm$^{-1}$ at 450 nm, and < 50 Mm$^{-1}$ at 630 nm) levels of $\sigma_{ext}$ through regulation of the dilution system. In the case of the external mixture of Aquadag and ammonium sulfate, the measurements were conducted under different SSA mixture conditions. The average SSA values determined with the A2S2 are ~0.71 (high SSA), ~0.67 (moderate SSA) and ~0.59 (low SSA) at 450 nm and ~0.66 (high SSA), ~0.65 (moderate SSA) and ~0.52 (low SSA) at 630 nm. The average normalised size distributions measured by SMPS are presented in Fig. 9 (normalised to the total aerosol number concentration for each SMPS scan). The size distribution results show that the median mobility diameter is smaller than 200 nm for all the groups, therefore the truncation correction for the A2S2 data can be ignored. The $\sigma_{ext}$ values and EAE measured by the A2S2 agree well with the results measured by two CAPS-PM$_{ex}$, as expected since they incorporate the same CAPS-based technique. This also confirms that our modified A2S2 monitor has equivalent performances than the currently available commercial CAPS monitors for the aerosol $\sigma_{ext}$ measurements. The A2S2 and NEPH instruments also show good agreement in measuring the $\sigma_{sca}$ and SAE across different conditions (differences within 10%). These results indicate that the measurements obtained from both instruments agree well and that the A2S2 provides consistent results for $\sigma_{sca}$ values under varying temperature and pressure conditions. On the other hand, the average difference of $\sigma_{abs}$ measured by AE33 and CAPS-PM$_{SSA}$ is within 10%. Large variance was observed for the AAE, which could be attributed to the variation in the contribution of Aquadag. Previous measurements performed by Foster et al. (2019) demonstrated that when Aquadag or standard BC loading drops, the variance of AAE becomes more pronounced from CAPS-PM$_{SSA}$ results. However, the average AAE derived from both A2S2 and AE33 was close to 0.4 which is the expected AAE of the standard Aquadag particles. Our results agree with the findings found in previous laboratory experiments involving the CAPS-PM$_{SSA}$, for example, where Perim De Faria et al. (2021) demonstrate that the CAPS-PM$_{SSA}$ can achieve

measurements of $\sigma_{ext}$ and $\sigma_{sca}$ at 630 nm with uncertainties within 10%, but the measurement of $\sigma_{abs}$ has uncertainties of 4% - 16%. Our results also agree well with the results from Weber et al. (2022) who found that the relative uncertainties for the $\sigma_{ext}$, $\sigma_{sca}$ and $\sigma_{abs}$ from CAPS-PM$_{SSA}$ at 450 nm and 630 nm are within 20% as suggested by Laj et al. (2020) for the ambient aerosol optical properties measurements. In addition, Corbin et al. (2022) also demonstrated good agreement between black carbon (BC) mass concentrations and $\sigma_{abs}$ from CAPS-PM$_{SSA}$ at 660 nm in an engine emission experiment.

## 3.2 Aircraft measurement results

### 3.2.1 Urban environment measurement results

The overview of the AOD values retrieved from AERONET observations at 440 nm and 675 nm during the ACROSS airborne flight campaign period as measured at a Paris urban site and at the rural site (in the Rambouillet Forest), and the AOD measured by the A2S2 as determined by integrating the altitude profile of $\sigma_{ext}$ at 450 nm and 630 nm are presented in Fig. 10(a). Fig. 10(b) and (c) display the SSA and SAE measured by the A2S2 at 450 nm and 630 nm within boundary layer for each flight. According to the AERONET reanalysis of AOD results over the same area as the aircraft flight operations, there are two periods during the campaign period: a heavily polluted period with AOD values up to 0.8 between 18[th] (Flight A025) and 23[rd] (Flight A028) June, and light pollution periods with AOD values around 0.2 for the remainder of the flights. The AERONET AOD values at 440 nm retrieved from the Paris urban site are higher than the results from the Rambouillet Forest site, whereas the AOD at 675 nm exhibits similar values at both sites. This could be attributed to the elevated concentration of non-refractory particulate matter in the urban area of Paris compared to the rural region. Comparing the AOD integrated from the altitude profiles of $\sigma_{ext}$ to the AERONET AOD results, the in-situ measured AOD result was lower than the AOD retrieved at Paris urban AERONET site due to lower pollution level but is close to the results at Rambouillet Forest site.

The average SSA within the boundary layer at 450 nm and 630 nm varied between 0.8 and 0.9 for the entire campaign, and the average SSA during the heavily polluted period (0.82 at 450 nm and 0.85 at 630 nm) is slightly lower than the average SSA during the lightly polluted period (0.87 at 450 nm and 0.90 at 630 nm). Due to the extremely low aerosol levels during the lightly polluted period, the $\sigma_{ext}$ and $\sigma_{sca}$ are close to the detection limits of the instrument which leads to relatively large uncertainties. The measured SSA observed in this study is close to the average SSA reported above the Greater London area in summer (0.89 and 0.88 at 467 nm and 652 nm, respectively) (Davies et al., 2019). During most flights, the average SAE values range between approximately 1 and 2. However, for Flights A025, A029, A032, and A033, the SAE dropped to around 1 due to the influence of larger-sized particles. The average AAE varied between 0 and 3, and this is potentially due to both the complicated emission sources and the low aerosol loading. Previous research has suggested that the stronger absorption at shorter wavelengths (average AAE > 1.5) could arise from either dust events (average SAE < 1.5) or a substantial contribution from brown carbon (BrC) (average SAE > 1.5) (Cappa et al., 2016). Our measurements indicate that dust particles contributed

to the overall aerosol loading during Flight A025 and A032, while BrC appears to have played a significant role in aerosol absorption during Flights A026, A027, A028 and A036.

The altitude profile results of Flights A025, A026, A028 and A030, and the aerosol optical properties above both the marine and continental background environments are presented are presented in Fig. 11. The results at 450 nm for A028 are not available due to the technical issues. The Flight A025 had the highest aerosol extinction and scattering coefficients among the four cases. The sharp decrease of SAE between 1500 and 2500 m indicates the presence of a dust layer, which contributed to the increase of aerosol extinction and scattering at both wavelengths. The SSA is also observed to decrease within the dust layer which could be caused by the mixture of dust with absorbing carbonaceous components and by the truncation correction errors. Due to the variation of boundary layer conditions, the profile of Flight A026 consists of two separate aerosol layers. One is within the boundary layer up to 1500 m, and the other one is at altitudes above 2000 m. The SAE at the upper layer is slightly less than the SAE at lower layer, and this indicates that the aerosols in the upper layer may be larger than the aerosols in the lower layer. For the Flights A028 and A030, the $\sigma_{ext}$ and $\sigma_{sca}$ values decreased with increasing altitude, and the relatively low $\sigma_{ext}$ indicates a relative clean background profile. The increase of SAE from ~1 to ~3 at altitudes above 1200 m for Flight A030 indicates that only fine mode particles are present at the upper level. The SSA of all the flights varies between 0.8 and 0.9 across the whole column, and there is a slight increase at the top for each flight indicating the reduction of the absorbing aerosols at the top.

### 3.2.1 Comparison of A2S2 and Nephelometer onboard the aircraft

The comparisons of measured $\sigma_{sca}$ by A2S2 and NEPH at 450 nm and 630 nm are presented in Fig. 12 and Fig. 13 respectively. For better comparison, the normalised probability density functions (PDF) of A2S2 and NEPH results for each flight are presented in the supplement. As discussed previously, there were dust (coarse mode particles) events present during Flights A025, A029, A032 and A032. During flight A025 the poorest agreement (12% differences) between the A2S2 and the NEPH among all the ACROSS flights was observed, and this was possibly attributed to the important contributions from dust particles, as indicated by the SAE and AAE results shown in Fig 10. This agrees with the lab results of the truncation characterisation that there are larger measurement errors for the A2S2 when measuring $\sigma_{sca}$ at larger particle sizes. After implementing the simple truncation corrections described previously, the discrepancies between the A2S2 and the NEPH results at both wavelengths are reduced to approximately 10%. For the airborne measurements during the lightly polluted periods, the uncertainty of A2S2 measured $\sigma_{sca}$ increases slightly because even a small change in baseline becomes significant as the $\sigma_{ext}$ approaches the detection limit. The relevance of this issue is particularly pronounced at the 450 nm wavelength due to the contribution from gas phase absorption as described previously. Overall, the A2S2 shows good agreement with the NEPH overall during the entire ACROSS campaign, and this validated that the A2S2 can adequately replace the NEPH to obtain reliable measurements of $\sigma_{sca}$ under polluted conditions.

**Summary and outlooks**

In this study, we introduced a customized version of airborne dual-wavelength SSA monitor based on the CAPS-PM$_{SSA}$ technique. As we configured it, the A2S2 can be used to conduct continuous measurements under low-pressure conditions down to 550 hPa with limited impact from high cabin temperatures. The truncation effect can be ignored for the particles with $D_p$ smaller than 200 nm, while for larger particles the truncation correction can be up to 20%. Following the truncation experiment, a truncation correction algorithm has been devised that operates in accordance with SAE principles, and the average truncation correction factors are 1.13 and 1.05 at the wavelengths of 450 nm and 630 nm, respectively. In order to achieve a balance between signal-to-noise ratio and the high-frequency demands of the ACROSS project, the airborne measurement data, originally captured at 1 Hz, has been integrated over a 10s period in this study. This adjustment significantly reduces the minimum detection limit (MDL) for $\sigma_{ext}$ and $\sigma_{sca}$ measurements by over 60% at 450 nm and by more than 80% at 630 nm. The aircraft measurements were conducted in environments with varying levels of anthropogenic pollution in northern France in the summer of 2022. The measurements include both the heavily polluted (AOD $\geq$ 0.5) and lightly polluted environments (AOD < 0.5). The SSA within the boundary layer as measured throughout the entire ACROSS flight campaign varied between ~0.8 and ~0.9, and the vertical structure of the aerosol optical properties varied. The SAE observed during the measurement period varied between 1 and 2, which indicates the contributions from different aerosol modes. For the fine mode particle dominated environment, A2S2 can provide continuous stable measurements with uncertainties of 10% compared to the truncation corrected NEPH measurements. The uncertainty increased when larger dust particles appeared but was still around 10% after implementing the simplified truncation correction method developed based on the PSL truncation characterisation results. However, both the irregular shape and the variation in refractive indices of the dust particles may cause large uncertainties in the spherical Mie-theory predictions. The refractive indices of dust particles typically range from 1.47 to 1.53 for the real part and 0.001 to 0.005 for the imaginary part in the visible range (Di Biagio et al., 2019). Therefore, it is difficult to accurately validate the truncation correction algorithms applied by either A2S2 or NEPH for these larger dust particles. There is a need for more comprehensive simulations and characterisations with morphology aware models like T-matrix to make accurate truncation error corrections, particularly under conditions involving super-micron dust particle events. Furthermore, as our truncation characterisation shows the uncertainties of A2S2 slight larger than those of the previous study of Onasch et al. (2015) using the CAPS-PM$_{SSA}$, the potential additional uncertainty source arising from the reflection of the glass tube within the cavity of CAPS-PM$_{SSA}$ may need to be addressed as well (Liu et al., 2018; Modini et al., 2021). Our laboratory and field measurement results validated the A2S2 as reliable for airborne measurements of aerosol scattering and extinction coefficients at both blue and red wavelengths under different ambient conditions. Laj et al. (2020) indicated that the uncertainty of in-situ measurement techniques for aerosol SSA characterization should be less than 20% to contribute effectively to climate studies. Our results demonstrate that the measurement uncertainties of A2S2 fall within the required

uncertainty ranges suggested by Laj et al. (2020). Therefore, the measurement results obtained from A2S2 can significantly
contribute to future climate modelling studies.

## Data availability

The data of the laboratory experiment are available through the Zendo: 10.5281/zenodo.10056220 (last accessed 31/10/2023).
Processed ACROSS flight campaign data and AERONET data are archived at ACROSS AERIS: https://across.aeris-data.fr/
(last accessed 26/10/2023).

## Author Contributions

PF designed the project; CY, PF, MC, KT, EP, MF, and VM performed the laboratory validation experiments; CY, PF, KT,
VM and CC performed the ACROSS airborne experiment; EP, KT, MC, and TB contributed to the instrument modification
and configuration onboard the ATR-42 aircraft;  TO and AF provided suggestions to the instrument modification; MF wrote
the new instrument software interface; CC designed the ACROSS-AO aircraft research; CY performed the data analysis; CY
and PF wrote the manuscript. All the co-authors contributed to the comments on the manuscript.

## Acknowledgment

Airborne data was obtained using the aircraft managed by Safire, the French facility for airborne research, an infrastructure of
the French National Center for Scientific Research (CNRS), Météo-France and the French National Center for Space Studies
(CNES). Chenjie Yu would like to acknowledge the Marie Skłodowska-Curie COFUND Paris Regional Postdoctoral
Fellowship program supported by the Paris Region. The authors acknowledge Aerodyne Inc. for their support in the
configuration of A2S2. A special thanks to Andreas Petzold (Forschungszentrum Jülich GmbH) for his assistance with the
early concept of the instrument.

## Financial support

The development of the A2S2 instrument was funded by the Institut National des Sciences de l'Univers du Centre National de
la Recherche Scientifique (CNRS INSU). Its airworthiness and integration on the ATR-42 was supported by the ACROSS
project, which benefits from French state aid (ANR – "Investissements d'avenir"); References: ANR-17-MPGA-0002 and
ANR-20-CE01-0010. This research also receives funding from the European Union's Horizon 2020 research and innovation
programme under the Marie Skłodowska-Curie grant agreement n° 945298.

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

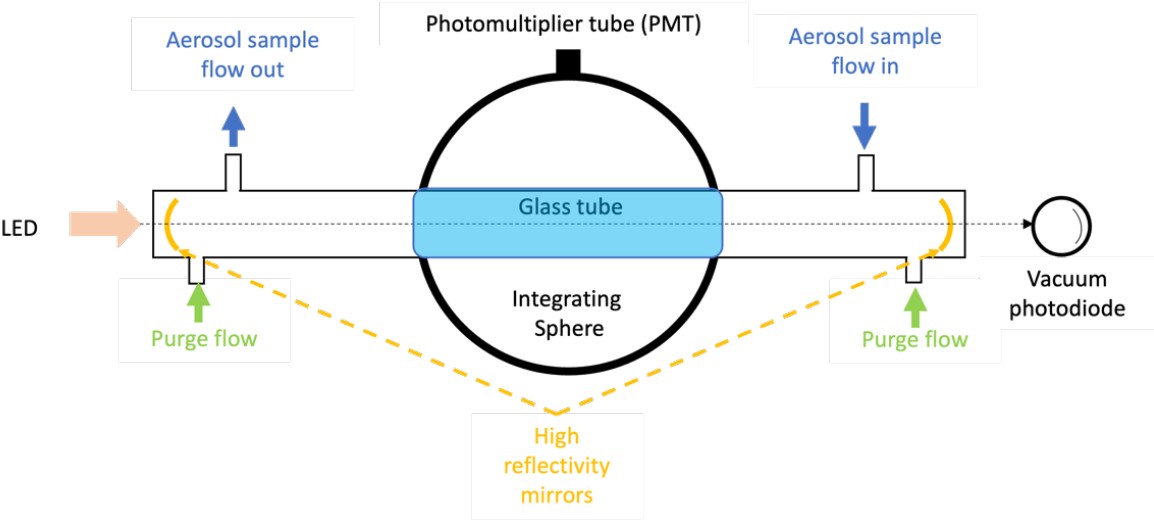


Figure 1 The diagram of the sample cell of the CAPS-PMssa (Adapted from Modini et al. (2021)).

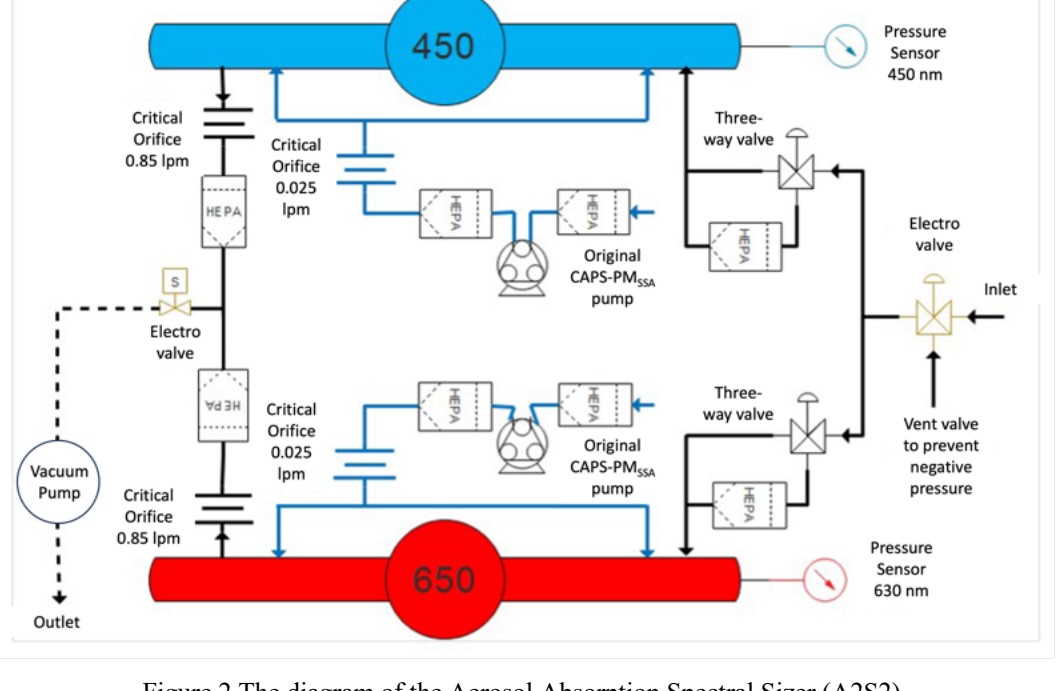

Figure 2 The diagram of the Aerosol Absorption Spectral Sizer (A2S2).

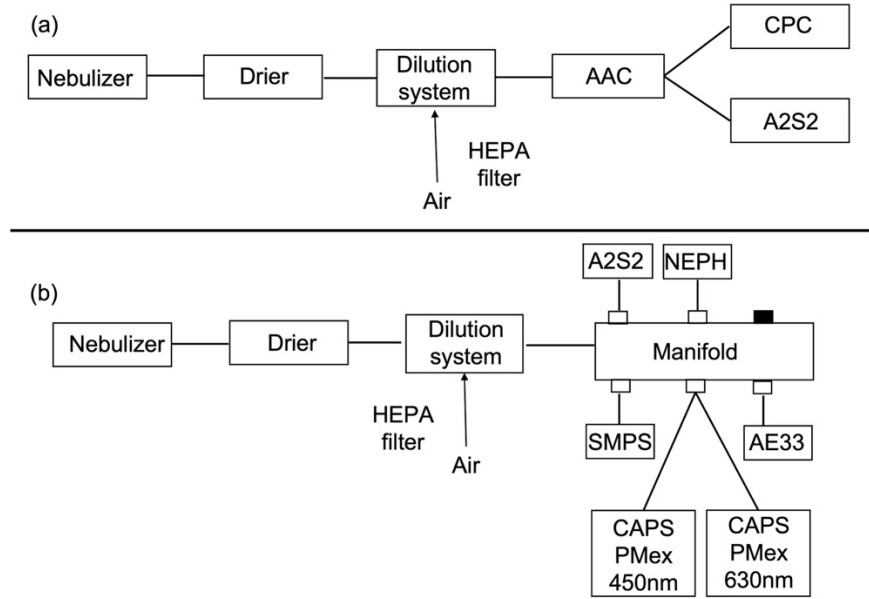

Figure 3 Instrument settings for (a) the truncation error characterisation; (b) the intercomparison study.

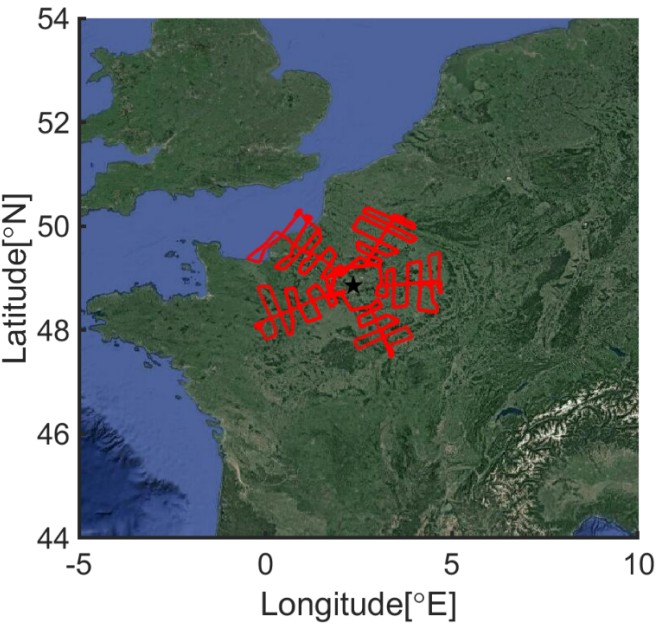

Figure 4. Flight patterns utilized during the ACROSS campaign. The red line shows the aircraft flight tracks, and the star symbol shows the location of Paris (from © Google Maps).

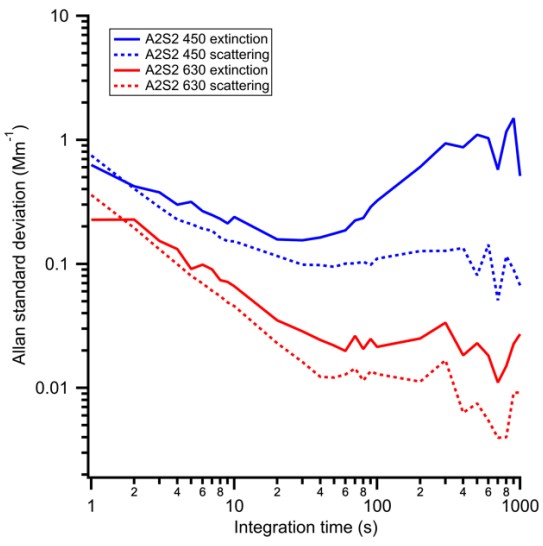

Figure 5 Allan standard deviation as a function of integration time at 450 nm and 630 nm.

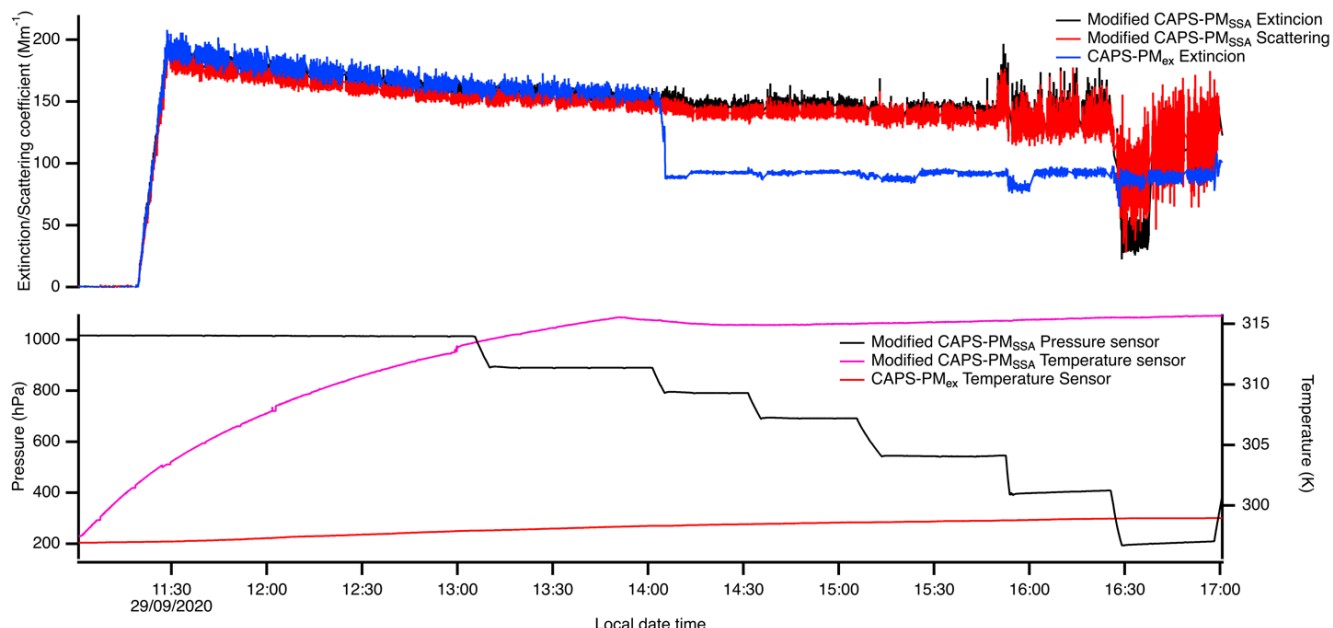

Figure 6. $\sigma_{ext}$ measured by CAPS-PM$_{ex}$ at 630 nm, and $\sigma_{ext}$ and $\sigma_{sca}$ measured by CAPS-PM$_{SSA}$ at several pressures; the temperature changes were applied to the CAPS-PM$_{SSA}$.

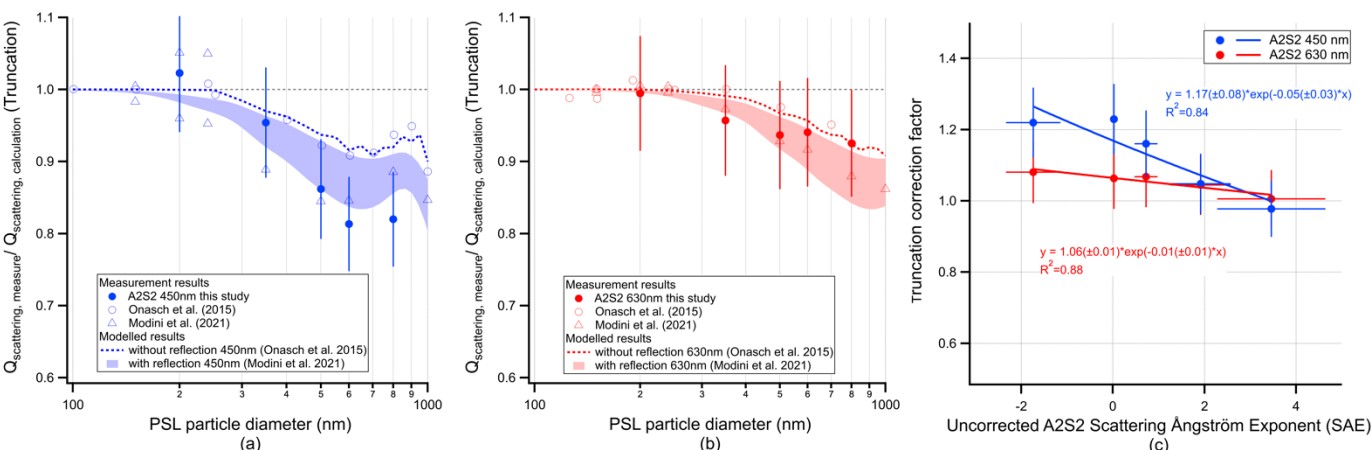

Figure 7. (a)(b) Measured and simulated truncation as a function of particle diameter using PSL particles at wavelengths of (a) 450 nm and (b) 630 nm. The simulated truncation is following the results in Onasch et al. (2015) and Modini et al. (2021). (c) Truncation correction factor as a function of measured uncorrected SAE. The error bars in the figure represents measurement precisions of the A2S2.

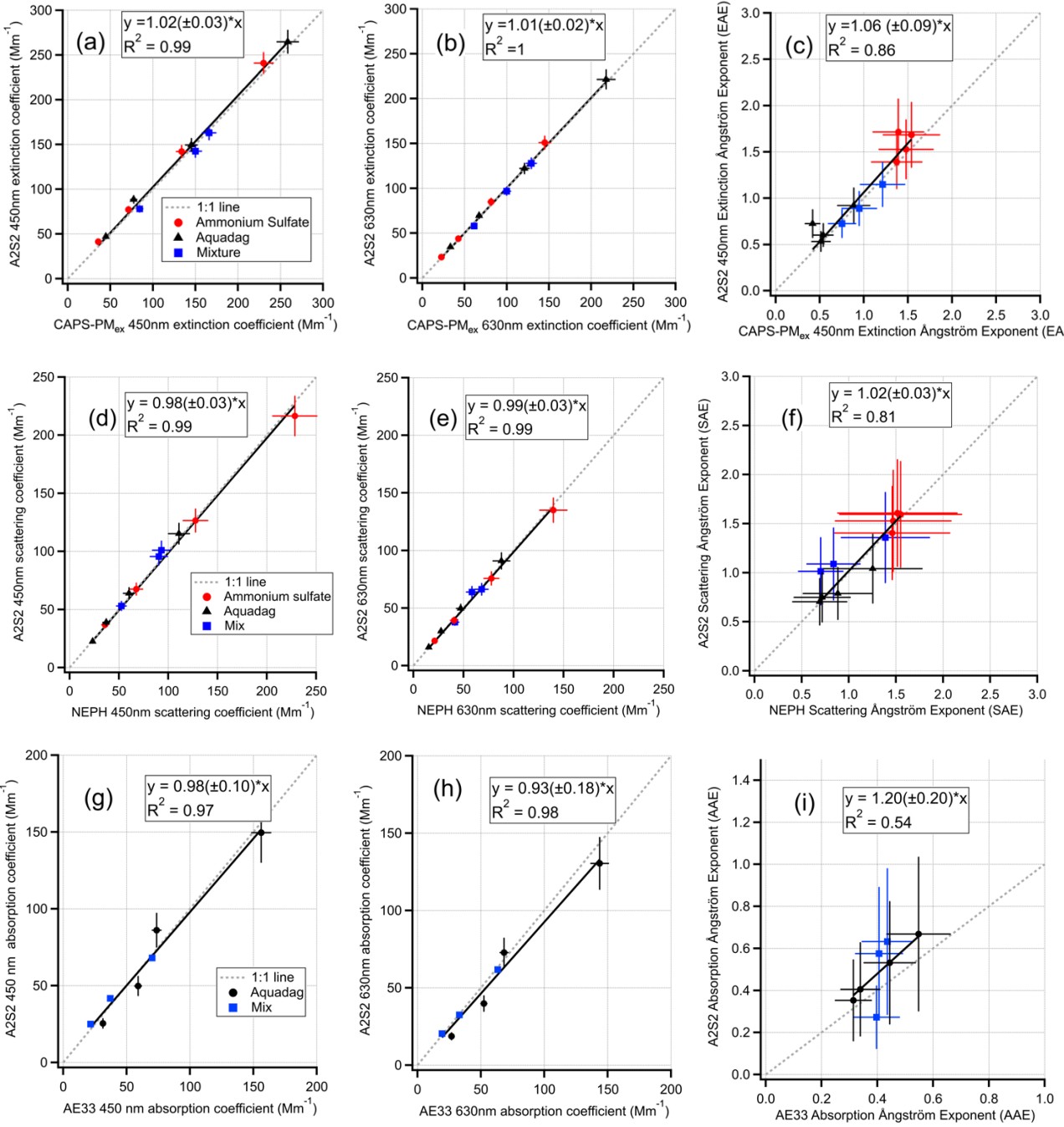

 Figure 8. Intercomparison (standard least square fitting) of A2S2 measurements with (a)(b)(c) extinction coefficients ($\sigma_{ext}$) and Extinction Angstrom Exponent (EAE) with CAPS-PM$_{ex}$; (d)(e)(f) scattering coefficients ($\sigma_{sca}$) and Scattering Angstrom Exponent (SAE) with NEPH; and (g)(h)(i) absorption coefficients ($\sigma_{abs}$) Absorption Angstrom Exponent (AAE) with AE33. The error bars in the figure represents instrument measurement precisions.


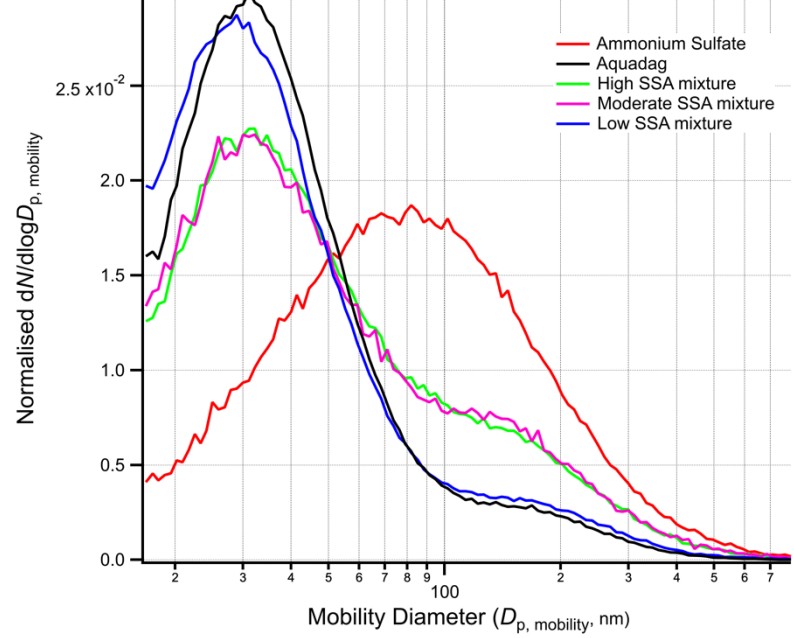

Figure 9. Average normalised size distribution of the aerosols for each intercomparison group as measured by SMPS.


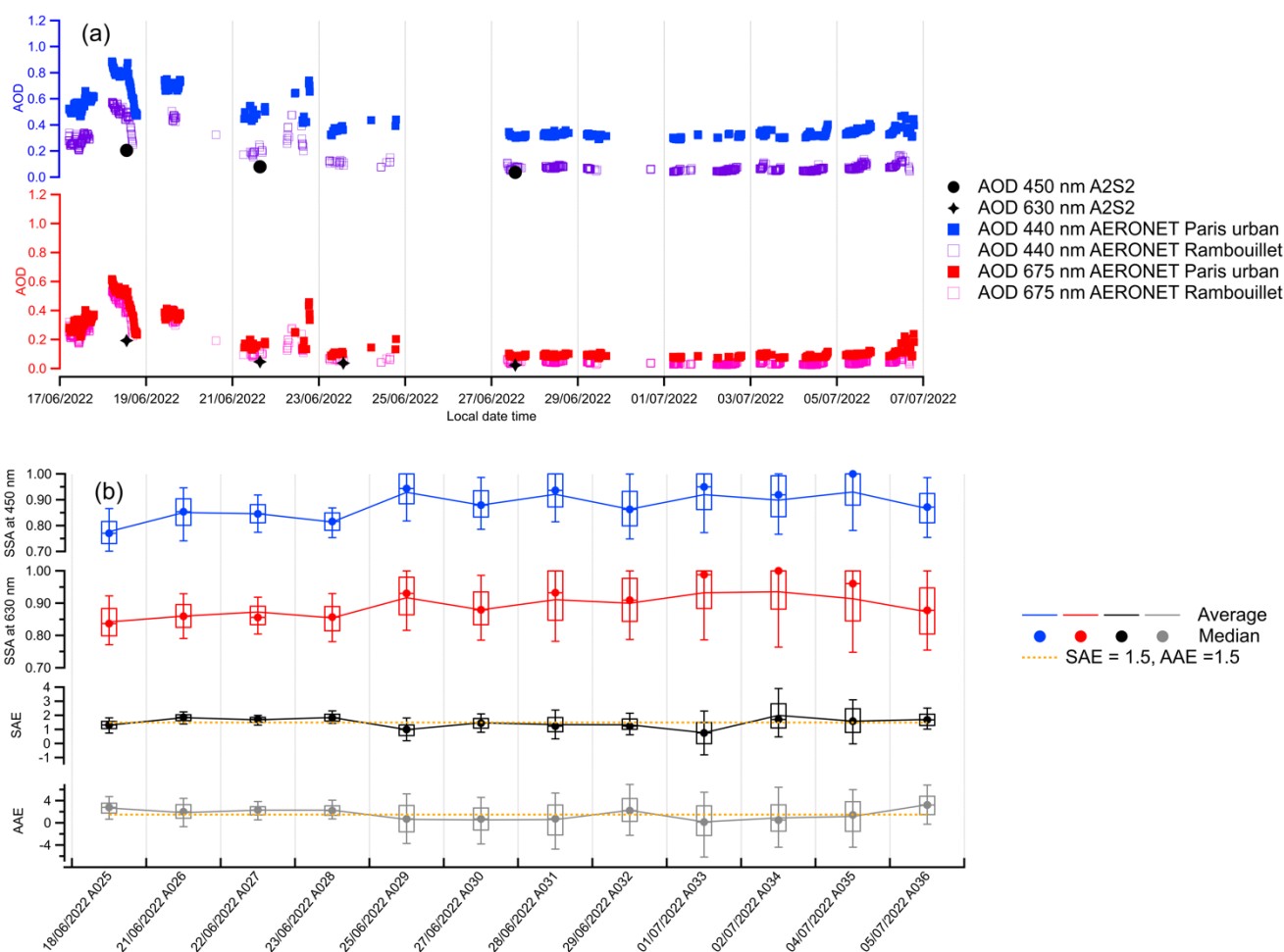

Figure 10. Time series of (a) Aerosol optical depth (AOD) from AERONET and A2S2 observation; (b) Aerosol SSA at 450 nm and 630 nm and SAE, and the box-and-whisker plots represent the average, 10th percentile, 25th percentile, median, 75th percentile and 90th percentile. The dashed yellow line indicates the SAE = 1.5 and AAE =1.5.


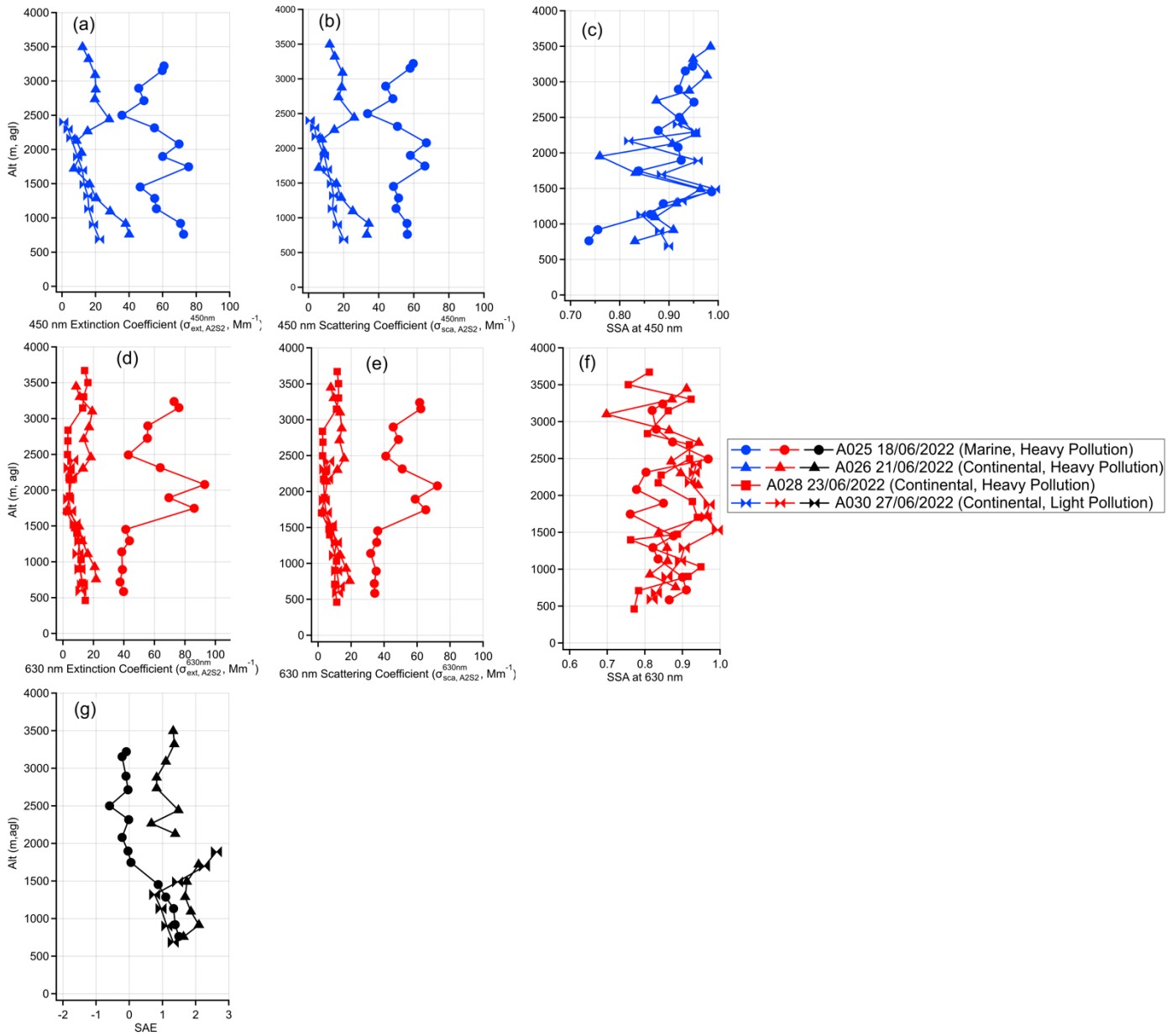

Figure 11. Altitude profile results for $\sigma_{ext}$, $\sigma_{sca}$ and SSA (a-c) at 450 nm and (d-f) at 630 nm and (g) SAE during the ACROSS campaign.

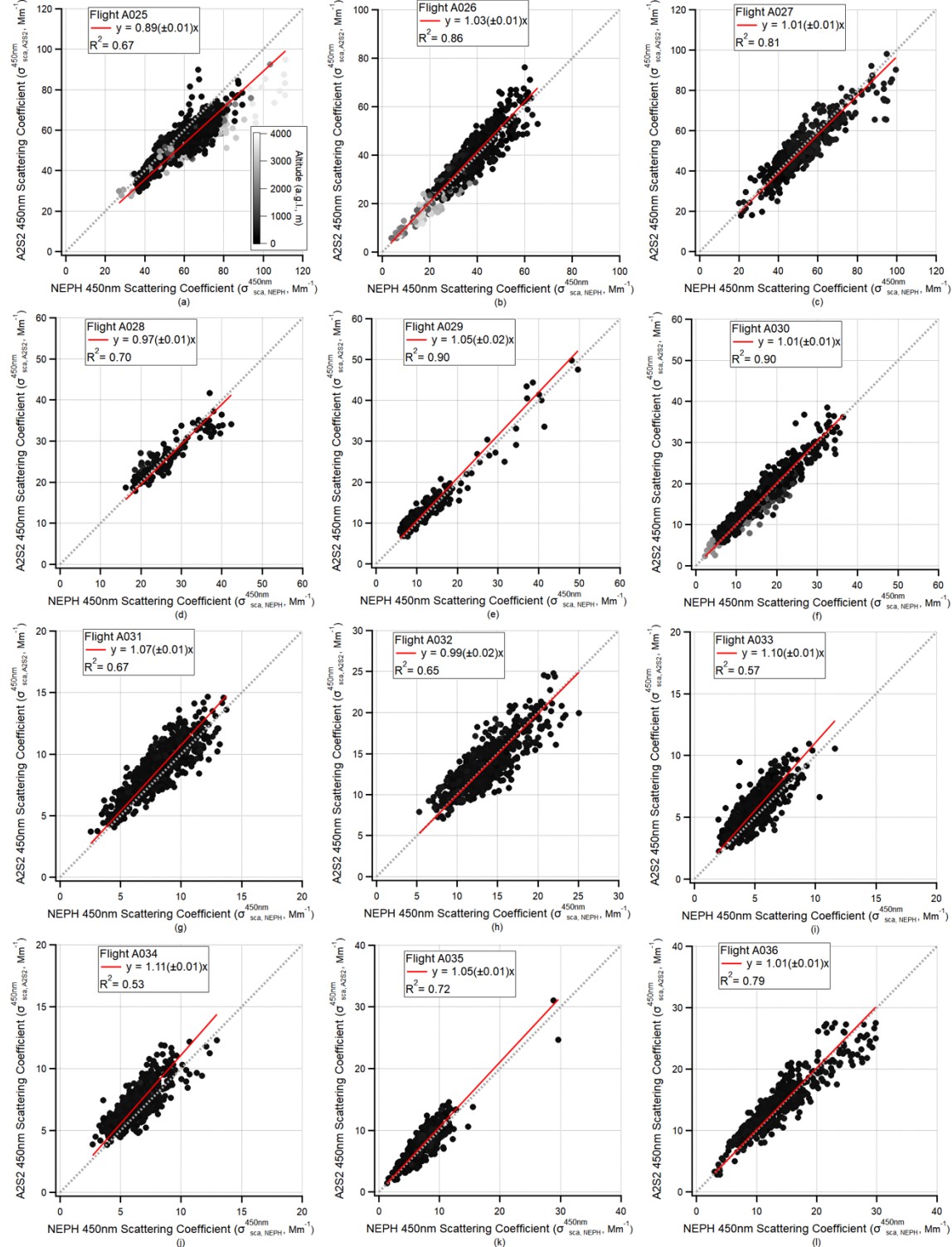


Figure 12. Comparison between A2S2 and NEPH at 450 nm of $\sigma_{sca}$ measurements for all the ACROSS flights.

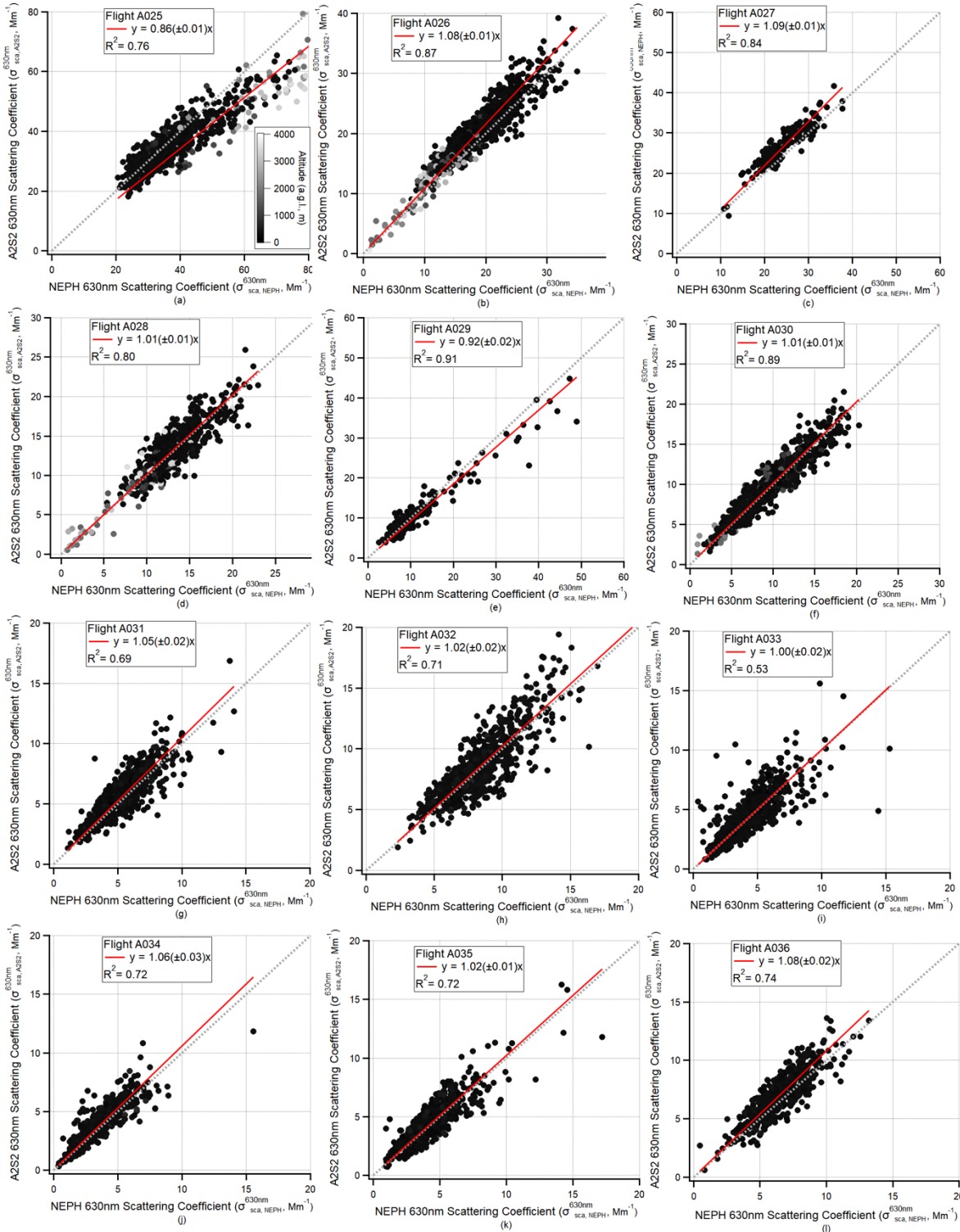

Figure 13. Comparison between A2S2 and NEPH at 630 nm of $\sigma_{sca}$ measurements for all the ACROSS flights.

| Instruments | Measurement parameters | Wavelengths (nm) | Original reference | Correction Algorithm Reference | Measurement uncertainty |
|---|---|---|---|---|---|
| A2S2 (Modified dual CAPS-PM$_{SSA}$) | $\sigma_{ext}$, $\sigma_{sca}$ | 450, 630 | Onasch et al. (2015) | - | 5% for $\sigma_{ext}$ 8% for $\sigma_{sca}$ |
| CAPS-PM$_{ex}$ | $\sigma_{ext}$ | 450, 630 | Massoli et al. (2010) | - | 5% |
| Nephelometer (NEPH) | $\sigma_{sca}$ | 450, 550, and 700 | Anderson et al. (1996) | Anderson and Ogren (1998) | 10% |
| Aethalometer (AE33) | $\sigma_{abs}$ | 370, 470, 520, 590, 660, 880, and 950 | Drinovec et al. (2015) | Bernardoni et al. (2021) | 5% |


Table 1 Instruments used in the intercomparison experiments performed in the laboratory.

| | | High level | Moderate level | Low level |
|---|---|---|---|---|
| Pure ammonium sulfate and pure Aquadag | 450 nm | $\sigma_{ext} > 200$ Mm$^{-1}$ | 100 Mm$^{-1} < \sigma_{ext} < 200$ Mm$^{-1}$ | $\sigma_{ext} < 50$ Mm$^{-1}$ |
| | 630 nm | $\sigma_{ext} > 150$ Mm$^{-1}$ | 50 Mm$^{-1} < \sigma_{ext} < 150$ Mm$^{-1}$ | $\sigma_{ext} < 50$ Mm$^{-1}$ |
| Aquadag and ammonium sulfate external mixture | 450 nm | SSA ~0.71 | SSA ~0.67 | SSA ~0.59 |
| | 630 nm | SSA ~0.66 | SSA ~0.65 | SSA ~0.52 |

Table 2. Expected $\sigma_{ext}$ and SSA at varying levels for the pure aerosol and aerosol mixture intercomparison measurement
studies, respectively.

| Integration time (s) | | 1 | 10 (applied in this study) | 30 |
|---|---|---|---|---|
| MDL for $\sigma_{ext}$ (Mm$^{-1}$) | 450 nm | 1.89 | 0.69 | 0.48 |
| | 630 nm | 0.69 | 0.21 | 0.09 |
| MDL for $\sigma_{sca}$ (Mm$^{-1}$) | 450 nm | 2.25 | 0.45 | 0.30 |
| | 630 nm | 1.08 | 0.12 | 0.06 |

Table 3. Minimum detection limit (MDL) of A2S2 at an integration time of 1s, 10s, and 30s.