# Peer review of "Characterisation of particle single scattering albedo with a modified airborne dual-wavelengths CAPS monitor"

_Atmospheric Measurement Techniques, 2023_

## Author Comment (AC1)

Firstly, we would like to thank both referees for their important comments, we have addressed all the comments below. The original comments from referees are in black, our replies are in blue and the changes in original manuscript are in red.

**Anonymous Referee #1**

This study provides a comprehensive characterization of the modified airborne dual-wavelength CAPS monitor ("A2S2") based the commercial version of Aerodyne CAPS-PMSSA. Through both laboratory investigations and field deployment, the modified instrument's performance has been validated. The results also reveal limitations in accurately characterizing the optical properties of larger particles, which could have implications for future studies employing CAPS-PMSSA. While overall this study fits the scope of AMT, the current manuscript appears to resemble a measurement report, lacking in-depth discussions regarding the obtained results:

Major comments:

1.  As this study includes some field deployment of their new instrument. However, it lacks some in-depth discussions. From a scientific point of view, what are the key findings from the observation results? According to a technical standpoint, how do the authors anticipate that their findings from the airborne measurements will implicate the future deployment of CAPS instruments?

We thank the reviewer for the comments. From a technical standpoint, the comments provided by Reviewer #1 align with that of Reviewer #2, and we express our gratitude to both of them for their valuable comments.

To address them, first of all we have added a discussion on the evaluation of the capability of the instrument to measure the spectral dependences of scattering, extinction and absorption (scattering angstrom exponent SAE, extinction angstrom exponent EAE and absorption angstrom exponent AAE, respectively). We have also worked out a truncation correction algorithm as a function of SAE based on our truncation experiment with PSL at different sizes. This algorithm will be crucial for improving scattering measurements from the A2S2 (dual CAPS-PM$_{SSA}$), particularly during dust events, in very helpful for the future deployments of the instrument. This information was also added to the revised abstract.

From the scientific standpoint, the field deployment was conducted in the framework of the ACROSS project, dedicated to the processes of formation of mixed anthropogenic/biogenic organic aerosols and the evaluation of their climate-relevant properties, optical in particular. In this respect, the new measurements by the dual CAPS system are key to the objectives of the project,

and their scientific exploitation and interpretation is the object of publication in progress. Henceforth, while we believe that a more detailed discussion of the relationship between aerosol compositions and aerosol optical properties is beyond the scope of this study, it is clear from the measurements presented in this paper that the mixed aerosols during the heat wave period of the campaign (flights A017 to A022) present significant absorption at both 450 and 630 nm, suggesting that light-absorbing organic aerosols should play a significant role. We will discuss this topic in more detail in our subsequent studies on this special issue, also relating our measurements to the concurrent results of the airborne AMS and TD-LIF instruments.

Nevertheless, we have added a discussion on the spectral dependences (response to Reviewer #2's major comment #15) and showed the evaluation of the dual CAPS-PM$_{SSA}$ under varying conditions, including during the dust event, when the SAE was close to 1 and the AAE greater than 1.5.

The optical-closure study with size-selected polystyrene latex (PSL) particles show that the truncation error of the A2S2 is negligible for particles with particle volume diameter ($D_p$) < 200 nm, while for the larger sub-micrometre particles, the measurement uncertainty of A2S2 increases but remains less than 20%. The average factors to correct the truncation error are 1.13 and 1.05 for 450 nm and 630 nm, respectively. A simplified truncation correction, dependent on the Scattering Ångström Exponent (SAE), was developed to rectify truncation errors of the future A2S2 field measurements data. The $\sigma_{ext}$ and $\sigma_{ext}$ measured by A2S2 shows good agreement with the concurrent measured results from the nephelometer and the CAPS-PM$_{ex}$ (Particle Extinction Monitor). The absorption coefficient $\sigma_{abs}$ derived through the extinction-minus-scattering (EMS) method by the A2S2 also corresponds with the results obtained from the aethalometer. The A2S2 was successfully deployed during an aircraft measurement campaign (ACROSS) conducted in the vicinity of Paris and the surrounding regions. The average SSA measured during the entire ACROSS flight campaign is 0.86 and 0.88 at 450 nm and 630 nm, respectively, suggesting that light-absorbing organic aerosols play a significant role. The average SAE and Absorption Ångström Exponent (AAE) varied due to measurements in various pollution conditions.

2. I am concerning about the baseline issue as described by Pfeifer et al. (2020). As the aircraft travels over the different backgrounds rapidly, will this issue persist in the airborne deployment of CAPS, even with more frequent baseline characterizations? The authors should give some more details regarding to this.

We appreciate the reviewer's query. However, as indicated by Pfeifer et al. (2020), the drift of baseline will only become an issue when the gas phase concentrations (for example NOx) change rapidly especially when the measurements conducted close to the fresh emission sources. During the ACROSS campaign, the gas phase background was without rapid variations, and the A2S2 was with high-frequency baseline chracterisations. We have revised the description here:

Nonetheless, the previous study by Pfeifer et al. (2020) shows that the variation of the gas phase $\sigma_{ext}$ baseline at 450 nm for the CAPS-PM$_{ex}$ may lead to an uncertainty up to around 0.8 Mm$^{-1}$ min$^{-1}$ for the ambient $\sigma_{ext}$ characterisations. To minimise the uncertainty of the baseline variation in CAPS-based instruments, a frequent baseline characterisation is needed. In this study, measurements were conducted at locations distant from the emission sources, and baseline values were measured every 2 min to reduce the influence from the background signal.

3. Page 10, Section 3.1.3, there are some other intercomparison studies for the CAPS-PMSSA (for example Perim de Faria et al., (2021)). The authors may improve their discussions by comparing their results with the previous studies.

We thank the reviewer's suggestions. We have revised the section including the intercomparison of AAE, SAE and EAE. In addition to the suggested references, we have also included the discussions with the results from Foster et al. (2019) and Weber et al. (2022), and the measurements from Corbin et al. (2022) here. Please refer to our response to Reviewer #2's specific comment #3 for the revised results and discussions.

Minor comments:

1. Font size is too small in some of the figures. Especially Figure 12 and Figure 13.

Accept, we have enlarged the font sizes here. Please refer to our revised manuscript for the improved graphs.

2. Figure 11, it is difficult to distinguish the symbols in the graph, suggest improving the graph.

Accept, we have revised the graphs here.

[Figure]

Figure 11. Altitude profile results for $\sigma_{ext}$, $\sigma_{sca}$ and SSA (a-c) at 450 nm and (d-f) at 630 nm and (g) SAE during the ACROSS campaign.

**Anonymous Referee #2**

This study uses a combined measurement package consisting of two commercially available CAPS PMSSA instruments. Laboratory inter-comparison results with several reference instruments were presented, and a flight analysis was performed using a nephelometer and remote sensing techniques for validation. The topic is within the scope of AMT and will attract many readers, but the current script lacks new insights and clarity.

Specific Questions and Comments

1. Please harmonize the script. Especially during introduction some statements were unreasonable

We have followed the reviewer's detailed suggestions to revise the related sections. Please refer to our responses in the Major and Minor comments for the revised paragraphs.

2. uncertainties were given as an absolute Value. Please give context for a relative value as well. There is no error analysis of the techniques being compared. Instead, variance the measurements is used for uncertainty. Please add this detail.

We thank the reviewer for the comments. We have included the absolute measurement uncertainties through referring to the previous studies in Table 1. In Fig. 8, we have also added the error bars based on measurement uncertainties. Uncertainties for the angström exponent have also been calculated based on error propagation (presented in our response to specific comment #2).

| Instruments | Measurement parameters | Wavelengths (nm) | Original reference | Correction Algorithm Reference | Measurement uncertainty |
|---|---|---|---|---|---|
| A2S2 (Modified dual CAPS-PM$_{SSA}$) | $\sigma_{ext}$, $\sigma_{sca}$ | 450, 630 | Onasch et al. (2015) | - | 5% for $\sigma_{ext}$ 8% for $\sigma_{sca}$ |
| CAPS-PM$_{ex}$ | $\sigma_{ext}$ | 450, 630 | Massoli et al. (2010) | - | 5% |
| Nephelometer (NEPH) | $\sigma_{sca}$ | 450, 550, and 700 | Anderson et al. (1996) | Anderson and Ogren (1998) | 10% |
| Aethalometer (AE33) | $\sigma_{abs}$ | 370, 470, 520, 590, 660, 880, and 950 | Drinovec et al. (2015) | Bernardoni et al. (2021) | 5% |

3. Please add an error analysis for the Angström exponent.

We have added the intercomparison of the Angstrom exponents of extinction and scattering (EAE and SAE, measured) as well as absorption (AAE, calculated) in the laboratory validation studies. The error propagation has also been calculated for the Angstrom exponent according to the Gaussian error propagation. The error bar here has been changed to present the measurement uncertainties. We have added the descriptions in Section 2.2:

For the A2S2, the uncertainty of the $\sigma_{abs}$ derived from EMS method is 13% according to Onasch et al. (2015) and Pfeifer et al. (2020). The uncertainties for the angström exponent are derived through the Gaussian error propagations (Weber et al., 2022):

$$\Delta xAE = \sqrt{\left(\frac{-1}{\ln(\lambda_1/\lambda_2) \cdot \sigma_{\lambda_1}} \cdot \Delta\sigma_{\lambda_1} \cdot \sigma_{\lambda_1}\right)^2 + \left(\frac{1}{\ln(\lambda_1/\lambda_2) \cdot \sigma_{\lambda_2}} \Delta\sigma_{\lambda_2} \cdot \sigma_{\lambda_2}\right)^2}$$

Where $xAE$ represents EAE, AAE or SAE; $\Delta\sigma$ represents the measurement uncertainty of the extinction, absorption and scattering coefficient measurement at certain wavelength.

[Figure]

[Figure]

Figure 8. Intercomparison (standard least square fitting) of A2S2 measurements with (a)(b)(c) extinction coefficients ($\sigma_{ext}$) and Extinction Angstrom Exponent (EAE) with CAPS-PM$_{ex}$; (c)(d) scattering coefficients ($\sigma_{sca}$) and Scattering Angstrom Exponent (SAE) with NEPH; and (e)(f) absorption coefficients ($\sigma_{abs}$) Absorption Angstrom Exponent (AAE) with AE33. The error bars in the figure represents instrument measurement precisions.

The $\sigma_{ext}$ values and EAE measured by the A2S2 agree well with the results measured by two CAPS-PM$_{ex}$, as expected since they incorporate the same CAPS-based technique. This also confirms that our modified A2S2 monitor has equivalent performances than the currently available commercial CAPS monitors for the aerosol $\sigma_{ext}$ measurements. The A2S2 and NEPH instruments also show good agreement in measuring the $\sigma_{sca}$ and SAE across different conditions (differences within 10%). These results indicate that the measurements obtained from both instruments agree well and that the A2S2 provides consistent results for $\sigma_{sca}$ values under varying temperature and pressure conditions. On the other hand, the average difference of $\sigma_{abs}$ measured by AE33 and CAPS-PM$_{SSA}$ is within 10%. Large variance was observed for the AAE, which could be attributed to the variation in the contribution of Aquadag. Previous measurements performed by Foster et al. (2019) demonstrated that when Aquadag or standard BC loading drops, the variance of AAE becomes more pronounced from CAPS-PM$_{SSA}$ results. However, the average AAE derived from both A2S2 and AE33 was close to 0.4 which is the expected AAE of the standard Aquadag particles. Our results agree with the findings found in previous laboratory experiments involving the CAPS-PM$_{SSA}$, for example, where Perim De Faria et al. (2021) demonstrate that the CAPS-PM$_{SSA}$ can achieve measurements of $\sigma_{ext}$ and $\sigma_{sca}$ at 630 nm with uncertainties within 10%, but the measurement of $\sigma_{abs}$ has uncertainties of 4% - 16%. Our results also agree well with the results from Weber et al. (2022) who found that the relative uncertainties for the $\sigma_{ext}$, $\sigma_{sca}$ and $\sigma_{abs}$ from CAPS-PM$_{SSA}$ at 450 nm and 630 nm are within 20% as suggested by Laj et al. (2020) for the ambient aerosol optical properties measurements. In addition, Corbin et al. (2022) also demonstrated good agreement between black carbon (BC) mass concentrations and $\sigma_{abs}$ from CAPS-PM$_{SSA}$ at 660 nm in an engine emission experiment.

4. Why is there no approach for the Extinction angstrom exponent or absorption angstrom exponent, since the combined instrument is capable of deriving those (Weber et al 2022)

We thank the reviewer's comment. For the laboratory analysis, we have added intercomparison for the Absorption Angstrom Exponent (AAE) and Extinction Angstrom Exponent (EAE). For the airborne measurements, we have added airborne measured AAE results to the Section 3.2. Incorporating the Major comment #15, we have added discussions for AAE during the airborne measurements. Please refer to our response to Specific comment #2 for the revised laboratory measurement results, and our response to Major comment #15 for the revised airborne measurement results. Due to the ultra-low $\sigma_{abs}$ value at high altitude during the campaign, the AAE is with large uncertainties at high altitude and thus the AAE is only available for the results within boundary layer.

5. Please add a statement or error analysis for using the Anderson and Ogren (1998) or Massoli (2009) algorithm on the nephelometer correction, since measurements were done at high polluted conditions.

We have added the statement for using the correction algorithms introduced by Anderson and Ogren (1998). Given the fact that the average and median SSA at both wavelengths for all the flights was over 0.7, the uncertainties of using Anderson and Ogren (1998) correction algorithm is within 5% according to the results of Massoli et al. (2009). We have added the discussions in Section 2.3:

The NEPH was calibrated with $CO_2$ and corrected for truncation error through the methods described by Anderson and Ogren (1998). Presented later in Fig. 10, the average and median SSA during the ACROSS airborne measurement period exceeded 0.7. Massoli et al. (2009) indicates that the uncertainties associated with applying the truncation correction method, as outlined by Anderson and Ogren (1998), to NEPH scattering are within 5% when the SSA is greater than 0.7.

6. "Aerosol Absorption Spectral Sizer" A2S2 seems like a great acronym (Popculture), Please explain why you use Sizer, since I would associate the term with a diameter.

It is certain that the acronym A2S2 might not be perfect but offers the great advantage of being concise and easy to write, say and remember, which will facilitate future scientific communication among colleagues, both in papers and conference presentations. Reviewer #2 is right by saying that the word "sizer" literally means "sorting by size", and indeed we slightly detoured the meaning to that of "an instrument that measures". Our English-speaking co-authors all agree with this acronym, indicating that the acronym is understandable and therefore deemed acceptable.

7. Please discuss if the combined instrument is within the requirements for modelling studies. Laj et al. 2020

We have added the suggested references in the Summary and outlook section:

Laj et al. (2020) indicated that the uncertainty of in-situ measurement techniques for aerosol SSA characterization should be less than 20% to contribute effectively to climate studies. Our results demonstrate that the measurement uncertainties of A2S2 fall within the required uncertainty ranges suggested by Laj et al. (2020). Therefore, the measurement results obtained from A2S2 can significantly contribute to future climate modeling studies.

8. Please overlook your graphs again!

We have revised the graphs by addressing all the referee's detailed comments. Please refer to our revised manuscript for the revised graphs.

Major Comments

1. Line 21 "550 hPa […] 315 K" Please add flight height and where the 315 was measured (inside the Instrument or Cabin…)

We appreciate the reviewer's suggestions. However, here we referred to the results from the laboratory validation experiments performed within the temperature- and pressure-controlled chamber environment. We have revised the sentence to be clearer:

The chamber experiments show that the A2S2 can perform measurements at sample pressures as low as 550 hPa and at sample temperatures as high as 315K. Based on the Allan analysis results, we have evaluated that the minimum detection limit of the measurements is  show that the measurements are with a limit accuracy of ~2 Mm$^{-1}$ at 450 nm and ~1 Mm$^{-1}$ at 630 nm for 1 Hz measurements of both scattering coefficients ($\sigma_{sca}$) and extinction coefficients ($\sigma_{ext}$).

2. Line 29 The conclusion to use the A2S2 to replace the nephelometer seem far fetched. At least three CAPS with different wavelengths would be needed to provide the same spectral information as a nephelometer. Secondly, the CAPS PMssa provide no backscattering coefficient. The Ecotech Aurora 3000 nephelometer seems to be a more reasonable TSI nephelometer replacement.

Thanks for the comments. In this study, we would like to highlight that CAPS-PM$_{SSA}$ can provide dual-wavelengths extinction and scattering measurements at the same time, and it has a compact size for the airborne measurements. We have revised the related conclusions in Line 31-32:

The results presented in this study indicate that the A2S2 instrument is reliable for measuring aerosol $\sigma_{sca}$ and $\sigma_{ext}$ at both blue and red wavelengths, and it stands as a viable substitute for future airborne evaluations of aerosol optical properties.

3. Line 76 "theoretically smaller truncation effect" Please add precise information. Additionally, impacts on ambient measurements, and refractive index on the truncation error were done for the nephelometer by Massoli et al. 2009. This seems to be a white spot for the CAPS PMssa scattering truncation correction.

We appreciate the reviewer's valuable comments. The integrated sphere is designed to minimize bias of the light collections with respect to angle. However, due to the of the complex cavity design within the CAPS-PM$_{SSA}$, the truncation angles of the CAPS-PM$_{SSA}$ cannot be determined precisely like the commercial nephelometers. We have revised the related paragraphs to avoid further misunderstood:

The CAPS-PM$_{SSA}$ incorporates an integrating sphere, which minimize the bias of the light collections with respect to angle when measuring $\sigma_{sca}$, and the $\sigma_{abs}$ can also be derived indirectly through the extinction-minus-scattering (EMS) method.

We agree with the reviewer that there is presently a deficiency in studies dedicated to thoroughly characterizing the truncation error as a function of refractive index for the CAPS-PM$_{SSA}$. Here in this study, we presented a simplified correction method based on the monodisperse of the standard pure scattering particles (PSL) at different particle sizes.

4. Line 85 "fast response" the CAPS PMssa needs about 7 seconds to flush the cavity. Is this still suitable for aircraft operation?

We appreciate the reviewer's concerns. Firstly, we have discussed in Section 2.1, Line 129 in our original manuscript about the set-up of the CAPS-PM$_{SSA}$ which includes 2-min measurement period and 1-min flushing and baseline characterisation period. For the measurement period, the CAPS-PM$_{SSA}$ will provide continues measurements at 1 Hz. Secondly, though CAPS-PM$_{SSA}$ needs time for the flush and baseline characterisation, it can provide measurements for both extinction and scattering with a time resolution of 1s, while NEPH can only provide scattering measurements. Comparing to the filter-based measurements such as the aethalometer with slower measurement frequency, the measurement frequency of CAPS-PM$_{SSA}$ significantly improved. Therefore, we consider that our modified A2S2 (CAPS-PM$_{SSA}$) still stands out for the aircraft operation.

5.  Line 94-95 "These Measurements […] "Please step down a notch. This statement is daring.

We appreciate the reviewer's comments. However, in fact, our ACROSS airborne measurement project results will be used as part of the development and validation for the IASI satellite retrieval products, and these measurement results will also contribute to the radiative forcing simulations with WRF-CHIMERE. The related studies will come later in the ACP/AMT journals as part of the ACROSS project outputs. We have revised the sentence to smooth the statement:

By providing with vertical profiles of climate-relevant properties such as aerosol single scattering albedo and aerosol optical depth, the A2S2 measurement results can play a role in helping the evaluation of the direct and semi-direct radiative effects in modelling studies and contribute to future advancements and validation of aerosol remote sensing products (Formenti et al., 2018).

6.  Line 104 "act as a nephelometer" No, integrating sphere and integrating nephelometer are not the same. Please rephrase

We have revised the related the description here:

Shown in Fig 1, an additional integrating sphere with an inner diameter of 10 cm is used to characterise the $\sigma_{sca}$.

7.  Line 122 Critical Orifices need a sufficient delta P to have a constant flow. Is this given at all Pressure ranges you operate?

We appreciate the reviewer's query. In a previous study (Perim De Faria et al., 2017) involving airborne CAPS measurements, it was found that the unstable flow at low pressure was attributed to limitations in the original double-head pumps. As presented in the A2S2 diagram in our original manuscript, the A2S2 is connected to the external pump (AVIRAD system pump) while the original double-head pumps are only used for purge flows. There is a mass flow controller (MFC) connected to the A2S2 regulate the maximum sample flow of 2 lpm, and this MFC also recorded the sample flow rate during the measurements. We have added the time series of sample flow rate at STP, altitudes and pressure for the flight A025 to the supplementary. Our airborne measurement results clearly illustrate that there is sufficient pressure difference to maintain the constant flow.

[Figure]

Figure S4. Time series of flow rate (STP) of A2S2, sample pressure, and altitude (a.g.l.) during Flight #25.

8. Line 142 The Allan variance gives you the lower detection limit with the highest precision at an certain integration time. Why is this not used for the measured flight data? (around 30 sec)

We appreciate the reviewer's query. For the ACROSS airborne measurement, there is a demand of high-frequency results to characterise the aerosol optical properties when the research aircraft intercepted with the urban plumes or performed altitude profile measurements. The limit of detection calculated from three times the Allan standard deviations at integration time of 1s, 10s and 30s are listed in Table below where there is a significant decrease of limit of detection from integration time of 1s to 10s, the decrease from 10s to 30s is limited. Furthermore, given that the ACROSS project is conducted mostly within boundary layer (at around 300 m a.g.l.), and the minimum value of $\sigma_{ext}$ and $\sigma_{sca}$ is much above the limit of detection for an integration time of 10s. Hence, considering all these factors, we believe that a 10s average in this study is adequate to achieve a balance between the high-frequency requirements essential for airborne characterisation and the desire for an improved signal-to-noise ratio. In addition, a 10s average also enables better joint case studies about the urban plumes with the other fast frequency measurement techniques used in this project (for example the single particle soot photometer (SP2, DMT) at 1Hz). We have included additional evidence in the discussions to provide support for our decision to opt for a 10s average in this study:

The minimum detection limit (MDL) involves a calculation derived from three times the Allan standard deviation, and the detection limit at an integration time of 1s, 10s and 30s is presented in Table 3. Our laboratory results show that, at a measurement frequency of 1 Hz, the MDL is 1.89 Mm$^{-1}$ for $\sigma_{ext}$ and 2.25 Mm$^{-1}$ for $\sigma_{sca}$ at 450 nm. At 630 nm, the MDL is 0.69 Mm$^{-1}$ for $\sigma_{ext}$ and 1.08 Mm$^{-1}$ for $\sigma_{sca}$. With the increase of integration time to 10s, the limits are reduced to 0.69 Mm$^{-1}$ and 0.21 Mm$^{-1}$ for $\sigma_{ext}$ at 450 and 630 nm, respectively, and to 0.45 Mm$^{-1}$ and 0.12 Mm$^{-1}$ for $\sigma_{sca}$ at 450 and 630 nm, respectively. The Allan analysis indicates that the limit of detection

reaches its minimum value at an integration time of 30s. Nevertheless, in the case of ACROSS airborne measurements, there is a requirement for high-frequency results to effectively characterize aerosol optical properties, especially during instances when the research aircraft intercepted with urban plumes or conducted altitude profile measurements. The minimum $\sigma_{ext}$ and $\sigma_{sca}$ observed during ACROSS campaign is both ~1.35 Mm$^{-1}$ at 450 nm, and both ~0.32 Mm$^{-1}$ at 630 nm. Hence, the detection limit achieved with a 10s integration time is deemed satisfactory to fulfill the requirements of the ACROSS project measurement.

| Integration time (s) | | 1 | 10 (applied in this study) | 30 |
|---|---|---|---|---|
| MDL for $\sigma_{ext}$ (Mm$^{-1}$) | 450 nm | 1.89 | 0.69 | 0.48 |
| | 630 nm | 0.69 | 0.21 | 0.09 |
| MDL for $\sigma_{sca}$ (Mm$^{-1}$) | 450 nm | 2.25 | 0.45 | 0.30 |
| | 630 nm | 1.08 | 0.12 | 0.06 |

Table 3. Minimum detection limit (MDL) of A2S2 at an integration time of 1s, 10s, and 30s.

9. Line 198 Were the CAPS PMssa calibrated with CO2 as well to check for the geometry correction factor? Modini et al. 2021

No, we haven't performed $CO_2$ calibrations. We followed the approach proposed by Onasch et al. (2015)by performing calibrations using PSL particles (described in Line 235 in our manuscript), and we would like to point out that Modini et al. (2021) similarly conducts non-absorbing particle calibrations for the CAPS-PM$_{SSA}$ exclusively. But we still thank the reviewer for the useful suggestions, we may take the $CO_2$ calibrations into consideration for the future CAPS-PM$_{SSA}$ experiments.

10. Line 249 How was the uncertainty determined?

We have revised the description, here we are referring to the limit of detection which is calculated from three times of Allan standard deviation. The revised discussion is presented in our response to Major comment #8.

11. Line 262 At lower pressures, the ratio of the flow between purge flow and sample flow may vary because critical orifices were used, causing disturbances

We appreciate the reviewer's query. We would like to point out that we have addressed the limitation of the original supplied diaphragm pump (as described by Perim De Faria et al. (2017)) through connecting the A2S2 to an external pump on the aircraft. As presented in our response to Comment #7, it is evident that the sample flow rate remains unaffected by variations in pressure,

thereby demonstrating that there is sufficient pressure difference to maintain the flow during the measurements. Thus, the application of critical orifices will not cause the disturbances.

12. Line 300 A correction function based on the SAE (scattering angstrom exponent) would be suitable instead of applying a factor as soon a certain threshold is passed

Thanks for the suggestions. We have worked out the truncation correction factor as a function of SAE. When measurement with SAE < 1, the correction function will be applied to the results. We have added the figure and we have revised the discussions:

[Figure]

Figure 7. (a)(b) Measured and simulated truncation as a function of particle diameter using PSL particles at wavelengths of (a) 450 nm and (b) 630 nm. The simulated truncation is following the results in Onasch et al. (2015) and Modini et al. (2021). (c) Truncation correction factor as a function of measured uncorrected SAE. The error bars in the figure represents measurement precisions of the A2S2.

Based on the characterisation results reported here, we introduce a simplified correction algorithm as a function of measured uncorrected Scattering Ångström Exponent (SAE) to apply to ambient measurement results. The correction function is presented in Fig. 7(c). Derived from the average truncation calculated above, the average correction factors are 1.13 and 1.05 at the wavelengths of 450 nm and 630 nm, respectively. Subsequently, the truncation is corrected based on the time-resolved measured uncorrected SAE between 450 nm and 630 nm observed by A2S2: when the SAE falls below 1, indicating the dominance of larger particles, the correction function is applied to the measurement results. Conversely, in situations dominated by fine particles (SAE > 1), there is no need to apply the correction due to the minimal truncation observed during characterisation experiments.

13. Line 323ff Be more precise and the uncertainty of the EMS (extinction minus scattering); may fit well even for pure aquadag aerosols at low aerosol load. Discuss with Weber et al 2022

We appreciate the reviewer's comments. We would like to claim that the $\sigma_{abs}$ derived by CAPS-PM$_{SSA}$ and through AE33 fits well in general (within 20% as suggested by Laj et al. (2020)) according to our results. But the low aerosol loading may lead to the variation of the AAE, this agrees with the findings from Foster et al. (2019). We have revised the discussions here, and we have also included the results from Weber et al. (2022) here. Please refer to our response to specific comment #3 for the revised discussions.

14. Line 356 High uncertainty of the angstrom exponent is most likely due the uncertainty of the CAPS scattering signal.

We appreciate the reviewer's comments. We have discussed in our original manuscript that the low aerosol loading during the lightly polluted period is close to the detection limit of the CAPS-PM$_{SSA}$ and thus leading to the uncertainty.

15. Line 364 mineral dust should have a different impact on the SSA for the different wavelengths since it is transparent at 630 nm but can absorb light at 450 nm. Please discuss. Can this be used for mineral dust determination?

We thank the reviewer's suggestions. Combined both SAE and AAE results, we can make some assumptions for the potential dust events or significant contribution from brown carbon (BrC). However, for more detailed discussions about the relationship between aerosol compositions and aerosol optical properties is beyond the scope of this study, and we will discuss this topic in more detail in our subsequent studies on this special issue. The revised results and discussions are as following:

During most flights, the average SAE values range between approximately 1 and 2. However, for Flights A025, A029, A032, and A033, the SAE dropped to around 1 due to the influence of larger-sized particles. The average AAE varied between 0 and 3, and this is potentially due to both the complicated emission sources and the low aerosol loading. Previous research has suggested that the stronger absorption at shorter wavelengths (average AAE > 1.5) could arise from either dust events (average SAE < 1.5) or a substantial contribution from brown carbon (BrC) (average SAE > 1.5) (Cappa et al., 2016). Our measurements indicate that dust particles contributed to the overall aerosol loading during Flight A025 and A032, while BrC appears to have played a significant role in aerosol absorption during Flights A026, A027, A028 and A036.

[Figure]

Figure 10. Time series of (a) Aerosol optical depth (AOD) from AERONET and A2S2 observation; (b) Aerosol SSA at 450 nm and 630 nm and SAE, and the box-and-whisker plots represent the average, 10th percentile, 25th percentile, median, 75th percentile and 90th percentile. The dashed yellow line indicates the SAE = 1.5 and AAE =1.5.

16. Line 383 a SAE based function might decrease the discrepancies even more.

We have followed the reviewer's suggestions to work out the SAE based truncation correction scheme. Please refer to our response to comment #12 for the details of the truncation correction function. The results have some minor improvements, but it will not influence our discussions. We have also combined the Minor comments from Reviewer #1 to enlarge the font sizes. The revised graphs are presented in our response to Reviewer #1's Minor comment #1.

17. Line 396 "70%" Please show this reduction, Please Discuss why 30 sec was not used (Allan variance determined) as integration time (10 sec might be the flush time of CAPS and NEPH)

We appreciate the reviewer's comments. Firstly, we would like to clarify that only the sampling periods data was included in the final results, and the flushing- and baseline-periods of CAPS have been excluded. NEPH was set to perform continues measurements. Integrations and intercomparisons are conducted exclusively for the CAPS and NEPH results obtained during the same time periods. Therefore, the flush time will not influence the integration calculations and intercomparison. Please refer to our response to Major comment #8 for the revised discussions why 10s is used here and the reduction of limitation. We have revised the summary here:

In order to achieve a balance between signal-to-noise ratio and the high-frequency demands of the ACROSS project, the airborne measurement data, originally captured at 1 Hz, has been integrated over a 10s period in this study. This adjustment significantly reduces the minimum detection limit (MDL) for $\sigma_{ext}$ and $\sigma_{sca}$ measurements by over 60% at 450 nm and by more than 80% at 630 nm.

18. Line 398 "clean background environment" might be different because of sea salt aerosols

We have revised the descriptions here:

The aircraft measurements were conducted in environments with varying levels of anthropogenic pollution in northern France in the summer of 2022.

19. Line 403 Please mention your truncation factor here as well

Accept. We have added the descriptions here:

The truncation effect can be ignored for the particles with $D_p$ smaller than 200 nm, while for larger particles the truncation correction can be up to 20%. Following the truncation experiment, a truncation correction algorithm has been devised that operates in accordance with SAE principles, and the average truncation correction factors are 1.13 and 1.05 at the wavelengths of 450 nm and 630 nm, respectively.

20. Line 406 Please discuss the refractive index of mineral dust as well (wavelength dependence of the complex refractive index)

We have revised the discussions here:

However, both the irregular shape and the variation in refractive indices of the dust particles may cause large uncertainties in the spherical Mie-theory predictions. The refractive indices of dust particles typically range from 1.47 to 1.53 for the real part and 0.001 to 0.005 for the imaginary part in the visible range. Therefore, it is difficult to accurately validate the truncation correction algorithms applied by either A2S2 or NEPH for these larger dust particles.

21. Line 414 "replacement for the NEPH on aircraft platforms" The nephelometer even works at lower pressure levels. Please rephrase!

We have revised the conclusions here. We would like to point out that we have performed the chamber validation for the performance of our modified A2S2 (CAPS-PMSSA) at low-pressure conditions. Combine the Reviewer's Minor comment #19, we have revised the statement here:

Our laboratory and field measurement results validated the A2S2 as reliable for airborne measurements of aerosol scattering and extinction coefficients under different ambient conditions.

22. Figure 1: Please redo the figure. Arrows do not point at the mirrors, the glass tube is missing

We have revised the graph here.

[Figure]

23. Figures: Please add grid-lines to all your graphs

We have added grid lines for part of the graphs. As we believe that too many elements could potentially detract from the readers' ability to interpret the data in certain graphs, we did not add grid lines to all the graphs. Please refer to our revised manuscripts for the revised graphs.

24. Figure 9 Unit for dn/dlogdp is 1/cm^3, not nm!

We have fixed the typo here. Here we use the normalised particle number distribution (normalised to the total aerosol number concentration for each SMPS scan), therefore there is no unit here.

[Figure]

Minor Comments

1. Line 16 "New pressure and temperature sensors" – Change to Additional or replaced

Accept. We have revised the description here:

Replaced pressure and temperature sensors and an additional flow control system were incorporated into the A2S2 for its utilization onboard research aircraft measuring within the troposphere.

2. Line 24 Truncation error. Please add your truncation factor up here

Accept. We have added the truncation error here:

According to the results of truncation error characterisation, a Scattering Ångström Exponent (SAE) based truncation correction algorithm has been applied to the ambient measurement data, and the average correction factors are 1.13 and 1.05 at the wavelengths of 450 nm and 630 nm, respectively.

3. Line 35 " and this is also known" please rephrase

We have rephrased the sentence here.

Atmospheric aerosols, particularly light-absorbing carbonaceous aerosols and mineral dust, play a significant role in global radiative transfer by scattering and absorbing solar radiation directly, a phenomenon referred to as the direct aerosol effect.

4. Line 57 Please a statement for the correction algorithm; Virkkula 2010

Accept. We have included the suggested references here:

For example, the filter-based absorption measurement method utilized by PSAP requires corrections to account for alterations in both scattering aerosol loading and aerosol transmissions (Virkkula et al., 2005; Virkkula, 2010).

5. Line 57 "relative slow" please rephrase and be more specific: not suitable for measurements on an aircraft due to the time resolution

Accept. We have revised the description here:

Moreover, the relatively slow measurement frequency of the filter-based measurement techniques makes them not ideal for the airborne measurements due to their slow time resolution, especially during altitude profiles.

6. Line 61 "limited range of scattering angles" please rephrase by adding the precise angles

We have added the description of angles within nephelometer here:

The nephelometer analyses the particle scattering intensity collected in a wide but limited range of scattering angles (7°- 170°), causing the loss of near forward and near backward scattering characterisation, a phenomenon commonly referred to as the truncation error.

7. Line 62 "truncation issue" change to truncation error

Accept. Please refer the revised description in our response to the Minor comment #6.

8. Line 63 "recent advancement" by citing a 20 year old paper…. This is not recent

We have revised our description here:

Technological advancements have allowed more precise direct measurements of the aerosol extinction coefficient ($\sigma_{ext}$).

9. Line 87 Please explain the purge flow

We appreciate the reviewer's suggestion, but we have explained the details of the sample and purge flow of the A2S2 in detail in the instrument section (Section 2.2, Line 121 – 123). To avoid duplication and confusion, we have revised the related descriptions here in the Introduction section as following:

an improved flow control system is required to maintain the instrument flows under the reduced-pressure conditions that are common during the airborne measurements.

10. Line 113 "into a single measurement monitor" into a combined package?

We have revised our description here:

we integrated two CAPS-PM$_{SSA}$ sample cells (450 and 630 nm, respectively) into a single measurement package

11. Line 123 "new […] sensors have been integrated" change to: existing sensors were replaced with

Accept. We have revised our description here:

Existing temperature and pressure sensors of CAPS-PM$_{SSA}$ were replaced with new temperature (DS18B20 by Maxim Integrated) and pressure (A-10-12719316 by WIKA) sensors, ensuring accurate monitoring of temperature and pressure to detect any leaking during airborne measurements.

12. Line 126 please add the uncertainty of the sensors

Accept. We have revised our description here:

The new pressure sensor has a measurement range from 0 to 40,000 hPa (within 0.5% uncertainties) while the range for the new temperature sensor is -55 ℃ – 125 ℃ ($\pm$0.5 ℃).

13. Line 129 Please add the explanation of the short duty cycle here

We have rephrased the sentence here to be clearer. The reason for the more frequent baseline characterisation setting is explained in Section 3.1.1 in our original manuscript.

The response time of A2S2 is 1 Hz, and it is programmed to carry out a continuous measurement phase for 2 min, which will be succeeded by a period of 1 min dedicated to cell flushing and establishing the baseline characteristics.

14. Line 173 "AAC can generate truly monodisperse" do you mean, without multi charged particles? Or is the GSD (global standard deviation) at a value of 1?

Here we indicate that the AAC classifies particles independently of their mass-to-charge ratio, thus the AAC monodisperse results are less influenced by particle compositions, morphologies, and sizes when compared to the electrostatic aerosol classifiers according to Tavakoli and Olfert (2013). We have revised the sentence here:

The AAC can generate monodisperse distributions of particles based on their aerodynamic sizes according to particle relaxation time without needing charging electrostatic elements. In contrast to electrostatic aerosol classifiers such as the differential mobility analyzer (DMA), the AAC can provide monodisperse results that are less affected by particle compositions, morphologies, and sizes.

15. Line 310-315 I suggest to add an table for all values for clarity

Accept. We have included the table as below:

| | | High level | Moderate level | Low level |
|---|---|---|---|---|
| Pure ammonium sulfate and pure Aquadag | 450 nm | $\sigma_{ext} > 200$ Mm$^{-1}$ | $100$ Mm$^{-1} < \sigma_{ext} < 200$ Mm$^{-1}$ | $\sigma_{ext} < 50$ Mm$^{-1}$ |
| | 630 nm | $\sigma_{ext} > 150$ Mm$^{-1}$ | $50$ Mm$^{-1} < \sigma_{ext} < 150$ Mm$^{-1}$ | $\sigma_{ext} < 50$ Mm$^{-1}$ |
| Aquadag and ammonium sulfate external mixture | 450 nm | SSA ~0.71 | SSA ~0.67 | SSA ~0.59 |
| | 630 nm | SSA ~0.66 | SSA ~0.65 | SSA ~0.52 |

Table 2. Expected $\sigma_{ext}$ and SSA at varying levels for the pure aerosol and aerosol mixture intercomparison measurement studies, respectively.

16. Line 330 Please at the wavelength you are comparing with. This paper mentions only the 630nm version

Accept. We have revised the related descriptions. Please refer to our response to Specific comment #3 for the revised descriptions.

17. Line 369 Please add the value of the SAE

We have revised the discussions:

The increase of SAE from ~1 to ~3 at altitudes above 1200 m for Flight A030 indicates that only fine mode particles are present at the upper level.

18. Line 377ff Please clarify "relative higher… relative poorer

We have revised the discussions here. We have moved the discussions for the dust particles to the section 3.2.1 following the suggestions:

During flight A025 the poorest agreement (12% differences) between the A2S2 and the NEPH among all the ACROSS flights was observed, and this was possibly attributed to the important contributions from dust particles, as indicated by the SAE and AAE results shown in Fig 10.

19. Line 388 "adequantely replace the NEPH" Please rephrase, so it become clear, that this is true, when only the scattering coefficient at two wavelengths is needed.

Accept. We have revised the summary here:

this validated that the A2S2 can obtain reliable measurements of $\sigma_{sca}$ at both blue and red wavelengths under polluted conditions.

**Reference**

[revised manuscript text omitted]